# ALLEVIATING SUBOPTIMALITY OF FLOW MAPS WITH IMPROVED SELF-DISTILLATION GUIDANCE

## ABSTRACT

Consistency-based approaches have been proposed for fast generative modeling, achieving competitive results compared to diffusion and flow matching models. However, these methods often rely on heuristics to mitigate training instability, which in turn limits reproducibility and scalability. To address this limitation, we propose the generalized flow map framework, unifying recent consistency-based methods under a common perspective. Within this framework, we investigate the suboptimality of existing approaches and identify two key factors for reproducibility: time-condition relaxation and marginal velocity guidance. To incorporate these, we leverage self-distillation to guide consistency models along the marginal velocity. We further propose *improved Self-Distillation (iSD)* by exploring the design space of flow maps, thereby reducing reliance on heuristics. Our formulation naturally extends to classifier-free guidance, achieving four-step generation with an FID of 11.06 on ImageNet $256 \times 256$. iSD shows qualitatively comparable results to prior few-step generative models, providing a theoretical and empirical foundation for reproducible consistency training.

## 1 INTRODUCTION

Diffusion (Ho et al., 2020; Song & Ermon, 2019a; 2020; Song et al., 2021) and flow matching models (Liu et al., 2023; Lipman et al., 2023) have achieved remarkable performance across a wide range of applications. This progress stems from flow-based modeling and multi-step inference, but is limited by multiple network evaluations during generation. To address this limitation, several works have explored improving sampling efficiency (Xiao et al., 2022; Salimans & Ho, 2022; Yin et al., 2024b;a; Zhou et al., 2024), but they rely on additional distillation stages or auxiliary networks to achieve fewer-step generation, thereby introducing additional training cost.

Consistency Models (Song et al., 2023) and its variants (Frans et al., 2025; Song & Dhariwal, 2024; Geng et al., 2025b; Lu & Song, 2025; Yang et al., 2024; Sun et al., 2025) have been proposed for training from scratch in few-step generation. Earlier studies on consistency models have often suffered from training instability, which led subsequent works to focus on stabilization by introducing various heuristics. However, the reliance on these complex techniques has reduced reproducibility.

In this work, we aim to develop a simplified and reproducible few-step generative model based on consistency methods. We begin by analyzing existing approaches: (i) we propose a generalized flow map framework that covers various design choices of recent consistency-based methods; (ii) we unify these approaches under our framework, providing a theoretical basis for analyzing suboptimality; and (iii) we show that *most of the recent methods do not guarantee convergence to the generator along the marginal velocity field, due to suboptimality and instability of their objectives.*

Motivated by these observations, we hypothesize that suboptimality and instability undermine the reproducibility of consistency training. From our unified perspective, we identify two key factors for reproducible training: time-condition relaxation and marginal velocity guidance. To incorporate these factors, we leverage our generalized flow map formulation with self-distillation. Some prior work (Issenhuth et al., 2025; Silvestri et al., 2025) have attempted to address these issues by reducing the loss variance, thereby resolving them indirectly. In contrast, self-distillation (Boffi et al., 2025a) was proposed to guarantee convergence to the marginal flow directly, but it relies on heuristics to stabilize training. By exploring the design space of flow map models, we propose *improved Self-*

*Distillation (iSD)*, which further reduces the reliance on heuristics. Moreover, we extend the iSD formulation to classifier-free guidance, achieving additional performance gains.

Our iSD demonstrates competitive performance against recent few-step generative models, while providing improved reproducibility. On ImageNet $256\times256$ (Deng et al., 2009), our model achieves an FID (Heusel et al., 2017) of 11.06 for four-step generation. The reproducibility of iSD is validated across multiple random initializations by measuring the standard deviation of FID, achieving 0.735. It demonstrates improved reproducibility compared to consistency training, as training can be performed from scratch with reduced heuristics.

**Contribution.** (i) We extend the flow map framework to cover various design choices, unifying recent consistency-based approaches within it (Sec. 3.2). (ii) We prove the suboptimal convergence and instability of gradient dynamics in recent consistency-based methods, showing that they undermine reproducibility and training stability (Sec. 3.3). (iii) To address these issues, we leverage the self-distillation, which guarantees convergence to the marginal velocity field (Sec. 4). (iv) We generalize the self-distillation to incorporate recent design choices and further extend it to classifier-free guidance, which we term *improved Self-Distillation* (iSD, Sec. 4). (v) We explore the design space and present the best choices for iSD. (Sec. 5).

## 2 RELATED WORK

**Diffusion and Flow Matching Models.** Diffusion models (Ho et al., 2020; Song & Ermon, 2019a; Song et al., 2021) and flow matching models (Albergo & Vanden-Eijnden, 2023; Albergo et al., 2023; Boffi et al., 2025b; Liu et al., 2023) are generative models that gradually transform a tractable noise distribution into the data distribution. These models have achieved remarkable progress in high-fidelity generation (Rombach et al., 2022; Podell et al., 2024; Peebles & Xie, 2023; Esser et al., 2024). However, their reliance on a multi-step sampling procedure requires substantial computational resources.

**Few-step Generation.** Several work have explored improving sampling efficiency of diffusion models (Salimans & Ho, 2022; Xiao et al., 2022; Rombach et al., 2022). These approaches aim to distill pretrained diffusion models into fewer-step generators, adopt GANs, or leverage VAEs to reduce input size. In parallel, distribution matching distillation methods (Yin et al., 2024b;a; Zhou et al., 2024) have been proposed to construct one-step generators by tracking the generator's score. However, both approaches rely on additional distillation stages or auxiliary networks, which increase training cost.

**Consistency Models.** Consistency Models (Song et al., 2023) are designed to predict a sample directly from any point along a flow trajectory. Both distillation and training methods have been proposed, but training from scratch is known to be unstable. Several studies have introduced heuristics to stabilize training, including initialization, improved objectives and progressive training schemes (Song & Dhariwal, 2024; Geng et al., 2025b; Lu

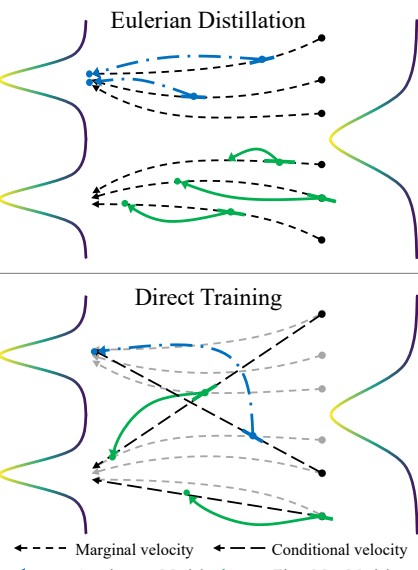

Figure 1: **Eulerian distillation and direct training of consistency models and flow map models.** Consistency models can be generalized into flow map models, which define a mapping between two points on the same trajectory. Eulerian distillation learns flow mappings along the marginal velocity, whereas direct training learns them along conditional velocity.

& Song, 2025). Other studies have identified the gap between distillation and training objectives (Issenhuth et al., 2025; Silvestri et al., 2025; Boffi et al., 2025b), which can lead to high loss variance and suboptimal convergence when training from scratch. To mitigate this, network-induced couplings have been introduced to reduce loss variance, addressing the issue indirectly. In contrast, self-distillation has been proposed to guarantee convergence directly, while dependent on heuristics.

**Unified Framework.** Recently, several studies have aimed to unify consistency models and flow matching. UCGM (Sun et al., 2025) introduced a framework that integrates both paradigms, but it does not account for the relaxed constraint of mapping points. More broadly, the flow map framework (Boffi et al., 2025b; Kim et al., 2024) presented a mathematical framework for consistency models, defining a model that learns *flow maps* as mappings between any two points on the same trajectory. However, it does not aim to unify recent work or explore their design spaces.

# 3 SUBOPTIMALITY OF DIRECT FLOW MAP MATCHING

In this section, we revisit the flow map framework (Boffi et al., 2025b; Kim et al., 2024) and extend it to interpret recent consistency-based approaches. We observe that most methods learn conditional velocity fields, which do not guarantee convergence to the marginal velocity field. Such suboptimal flow maps may lead to trajectory crossings, non-injective mappings, or severe reproducibility issues.

## 3.1 PRELIMINARIES

Given a training dataset $X$ with underlying distribution $p_X$, flow matching models are trained to match the velocity fields of continuous flows, starting from a tractable distribution $p_Z$. Prior work constructs such flows via an interpolation, $x_t = \alpha_t x + \sigma_t z$, where $x \sim p_X$ and $z \sim p_Z$. A linear interpolation $x_t = (1-t)x + tz$ for $t \in [0,1]$ (Liu et al., 2023; Lipman et al., 2023) and a trigonometric interpolation $x_t = \cos(t)x + \sin(t)z$ for $t \in [0, \pi/2]$ (Albergo & Vanden-Eijnden, 2023; Albergo et al., 2023; Lu & Song, 2025) are the widely adopted choices.

We assume that $\alpha_t$ and $\sigma_t$ are monotone with boundary conditions $\alpha_0 = \sigma_T = 1$ and $\alpha_T = \sigma_0 = 0$ for $t \in [0,T]$. Both are continuous and have bounded first- and second-order derivatives. Under this assumption, the marginal distribution induced by the flow, $\rho_t$, satisfies $\rho_0 = p_X$ and $\rho_T = p_Z$.

To ensure convergence of the consistency objective in the subsequent discussion, we propose an additional assumption that $\alpha_t \sigma_t' - \sigma_t \alpha_t' = \nu \neq 0$ for all $t \in [0,T]$ where $\nu$ is constant (see Appendix A.4). Notably, both linear and trigonometric interpolations satisfy this condition with $\nu = 1$.

With the constructed flow, the flow matching models optimize the squared error between the conditional velocity $v_t(x_t|x)$ and a parameterized velocity network $F_\theta$:

$$\mathcal{L}_{\text{CFM}} = \mathbb{E}_{x \sim p_X, z \sim p_Z, t \sim \mathcal{U}[0,1]} \left[ \|F_\theta(x_t; t) - v_t(x_t|x)\|_2^2 \right], \quad v_t(x_t|x) = \alpha_t' x + \sigma_t' z \quad (1)$$

Conditional flow matching $\mathcal{L}_{\text{CFM}}$ converges to the flow induced by the marginal velocity $v_t^*(x_t) = \mathbb{E}_{x|x_t}[v(x_t|x)]$. If $v_t^*(x)$ is Lipschitz continuous in both $t$ and $x$, the ODE $dx_t = v_t^*(x_t)dt$ has a unique solution and satisfies the continuity equation for $\rho_t$ (Lipman et al., 2023).

However, we observe that diffusion and flow matching models can suffer from *mean collapse*, in which one-step samples collapse to the mean of the data distribution (proof in Appendix A.1).

## 3.2 GENERALIZED FLOW MAP MATCHING

We begin by introducing a flow map with Eulerian distillation following prior work, and then generalize these formulations to unify recent consistency-based methods. Within our extended framework, we interpret recent methods and provide a theoretical basis for analyzing their suboptimality.

**Flow Map.** From the flows defined by an interpolation, our goal is to draw samples from the target distribution in a few sampling steps. To achieve this, we define a *flow map* as a mapping between two points $x_t$ and $x_s$ ($s < t$) on the same trajectory, using the marginal velocity $v_t^*(x_t)$ as follows:

$$f_{t,s}(x_t) = x_t + \int_t^s v_\tau^*(x_\tau)d\tau. \quad (2)$$

If the marginal velocity is Lipschitz continuous, the flow map is well-defined, injective, and thus *free from the mean collapse problem* (proof in Appendix A.2).

**Training Flow Map.** Since the flow map is defined as an integral, direct supervision from scratch is challenging. Assuming a teacher flow matching network $F_\Phi$, we can generate training data using an ODE solver, enabling direct supervision in a distillation manner. This procedure reduces to the

one-step distillation of rectified flows (Liu et al., 2023) for fixed $t = 1$ and $s = 0$. However, it requires a data generation process with substantial computational resources.

To address this limitation, many recent studies adopt consistency training (Song et al., 2023), which can be derived from the Eulerian equation (proof in Appendix A.3; see also Boffi et al. (2025b)).

**Proposition 3.1.** *(Eulerian Equation) Given the flow induced by a Lipschitz continuous marginal velocity $v_t^*(x_t)$, the flow map $f_{t,s}$ is the unique solution of the Eulerian equation:*

$$\partial_t f_{t,s}(x) + v_t^*(x_t) \cdot \nabla_x f_{t,s}(x) = 0 \tag{3}$$

*when $f$ is continuous in $x, t, s$, Lipschitz continuous in $x$, and satisfies the boundary $f_{t,t}(x) = x$.*

To facilitate training of a flow map network $f_\theta$ using the Eulerian equation, the training objective can be formulated as a squared minimization problem, referred to as *Eulerian distillation*:

$$\mathcal{L}_{\text{ED}} = \mathbb{E}_{x,z,t,s} \left[ \|\partial_t f_\theta(x_t; t, s) + v_t^*(x_t) \cdot \nabla_x f_\theta(x_t; t, s)\|_2^2 \right] \tag{4}$$

When $s = 0$, Eulerian distillation reduces in value to continuous-time consistency training objective (Song et al., 2023) (details in Appendix A.5). However, their gradient dynamics differ, which affects the training stability, as further discussed in a later section (Prop. 3.4).

Since the objective involves a Jacobian-vector product, its optimization requires computing a Hessian, introducing computational overhead. To address this, Lu & Song (2025) reformulated the consistency training objective with a stop-gradient operation, while keeping the gradients identical:

$$\mathcal{L}_{\text{CT}} = \mathbb{E}_{x,z,t} \left[ \left\| f_\theta(x_t; t) - f_{\theta^-}(x_t; t) + \frac{df_{\theta^-}(x_t; t)}{dt} \right\|_2^2 \right] \tag{5}$$

where $f_{\theta^-}$ denotes the gradient-detached network. Using Eq. 5, the JVP operation is detached from the gradient flow, and the update step requires two backward passes but avoids Hessian computation.

**Generalized Flow Map.** To interpret recent approaches as flow maps, *we propose a generalized formulation of the network $f_\theta$ as a one-step Euler solution* with a pseudo-velocity network $F_\theta$:

$$f_\theta(x_t; t, s) = \nu^{-1}(A'_{t,s} x_t - A_{t,s} F_\theta(x_t; t, s)), \quad A_{t,s} = \sigma_t \alpha_s - \sigma_s \alpha_t \tag{6}$$

This formulation satisfies the boundary condition of Prop. 3.1. For a linear trajectory, it reduces to $f_\theta(x_t; t, s) = x_t + (s - t)F_\theta(x_t; t)$, widely adopted in flow map studies (Geng et al., 2025a; Boffi et al., 2025b; Sabour et al., 2025). At $s = 0$, it simplifies to $f_\theta(x_t; t) = x_t - tF_\theta(x_t; t)$, the common setting in consistency models (Yang et al., 2024; Sun et al., 2025). For a trigonometric trajectory with $s = 0$, we obtain $f_\theta(x_t; t) = \cos(t)x - \sin(t)F_\theta(x_t; t)$, as introduced by Lu & Song (2025).

By generalizing the formulations of flow maps, we propose instantiations of these methods under a unified perspective (proof in Appendix A.5).

**Proposition 3.2.** *(Interpretation of Recent Methods) Recent consistency-based methods can be interpreted as instances of the flow map framework, trained with the generalized Eulerian equation:*

$$\partial_t f_\theta(x_t; t, s) + \tau_t(x_t, x) \cdot \nabla_x f_\theta(x_t; t, s) = 0 \tag{7}$$

*where $x_t$ is determined by the Interpolant, $\tau_t$ by the Trajectory, and the constraints of $t, s$ by Timestep, as summarized in Tab. 1.*

To be consistent with the generalized Eulerian equation, we extend Eulerian distillation and consistency training objective for $f_\theta(x_t; t, s) = f_{t,s}^\theta(x_t)$. Since the guiding trajectory $\tau_t$ is not generally Lipschitz continuous, it no longer guarantees the convergence to the marginal flow map.

$$\mathcal{L}_{\text{ED}} = \mathbb{E}_{x,z,t,s} \left[ \left\| \partial_t f_{t,s}^\theta(x_t) + \tau_t(x_t, x) \cdot \nabla_x f_{t,s}^\theta(x_t) \right\|_2^2 \right] \tag{8}$$

$$\mathcal{L}_{\text{CT}} = \mathbb{E}_{x,z,t,s} \left[ \left\| f_{t,s}^\theta(x_t) - f_{t,s}^{\theta^-}(x_t) + \partial_t f_{t,s}^{\theta^-}(x_t) + \tau_t(x_t, x) \cdot \nabla_x f_{t,s}^{\theta^-}(x_t) \right\|_2^2 \right] \tag{9}$$

We observe several design choices from these instantiations. Prior studies have focused on linear and trigonometric interpolations. The guiding trajectory typically follows either pretrained velocity networks or the conditional velocity. Consistency models usually fix $s = 0$, whereas other flow map models relax it to $s < t$. Some models compute the JVP directly using `torch.func.jvp`, while others approximate it as $df_\theta(x_t; t)/dt \approx [f_\theta(x_{t+\epsilon}; t + \epsilon) - f_\theta(x_{t-\epsilon}; t - \epsilon)]/(2\epsilon)$.

We explore these design spaces in the experimental section, and select the best settings: (i) trigonometric interpolation, (ii) approximated marginal velocity, (iii) $s < t$, and (iv) approximated JVP.

Table 1: **Instantiation of the flow map framework** for each consistency-based generative model.

| Model | Interpolant $x_t$ | Trajectory $\tau_t$ | Timestep | JVP Type | Loss | Note |
|---|---|---|---|---|---|---|
| **Distillation-based Methods** | | | | | | |
| FMM-EMD (Boffi et al., 2025b) | Linear | $F_\Phi(x_t; t)$ | $s < t$ | Exact | $\mathcal{L}_{\text{ED}}$ | Teacher $F_\Phi(x_t; t)$ |
| AYF-EMD (Sabour et al., 2025) | Linear | $F_\Phi(x_t; t)$ | $s < t$ | Exact | $\mathcal{L}_{\text{CT}}$ | |
| sCD (Lu & Song, 2025) | Trigonometric | $F_\Phi(x_t; t)$ | $s = 0$ | Exact | $\mathcal{L}_{\text{CT}}$ | |
| **Consistency Training Methods** | | | | | | |
| MeanFlow (Geng et al., 2025a) | Linear | $v_t(x_t\|x)$ | $s < t$ | Exact | $\mathcal{L}_{\text{CT}}$ | |
| ConsistencyFM (Yang et al., 2024) | Linear | $v_t(x_t\|x)$ | $s = 0$ | Approx | $\mathcal{L}_{\text{CT}}$ | |
| sCT (Lu & Song, 2025) | Trigonometric | $v_t(x_t\|x)$ | $s = 0$ | Exact | $\mathcal{L}_{\text{CT}}$ | |
| UCGM (Sun et al., 2025) | Arbitrary | $v_t(x_t\|x)$ | $s = 0$ | Approx | $\mathcal{L}_{\text{CT}}$ | |
| Shortcut Model (Frans et al., 2025) | Linear | $F_\theta(x_t; t, t)$ | $s = t + d$ | Approx | $\mathcal{L}_{\text{CTM}}$ | $d \in [-2^{-1}, -2^{-7}]$ |
| **Self-Distillation Methods** | | | | | | |
| Self-Distillation (Boffi et al., 2025a) | Linear | $F_\theta(x_t; t, t)$ | $s < t$ | Exact | $\mathcal{L}_{\text{SD}}$ | |
| **improved Self-Distillation (Ours)** | Arbitrary | $F_\theta(x_t; t, t)$ | $s < t$ | Approx | $\mathcal{L}_{\text{SD-R}}$ | |

## 3.3 SUBOPTIMALITY AND INSTABILITY

Most consistency training approaches learn flow maps guided by the conditional velocity. We denote Eulerian distillation under conditional velocity guidance as *direct training*.

$$\mathcal{L}_{\text{DT}} = \mathbb{E}_{x,z,t,s} \left[ \|\partial_t f_\theta(x_t; t, s) + v(x_t|x) \cdot \nabla_x f_\theta(x_t; t, s)\|_2^2 \right] \tag{10}$$

However, under the assumption in Prop. 3.1, the direct training does not guarantee convergence to the flow map along the marginal velocity, due to the gap between conditional and marginal velocity guidance (proof in Appendix A.6; cf. Boffi et al. (2025b)).

**Proposition 3.3.** *(Suboptimality of direct training) The gap between Eulerian distillation $\mathcal{L}_{\text{ED}}$ along the marginal velocity and direct training $\mathcal{L}_{\text{DT}}$ is given by*

$$\mathcal{L}_{\text{DT}} - \mathcal{L}_{\text{ED}} = \mathbb{E}_{x,z,t,s} \left[ \text{Var}_{x|x_t}[\Delta v \cdot \nabla_x f_\theta(x_t; t, s)] \right] \tag{11}$$

*where $\Delta v = v(x_t|x) - v_t^*(x_t)$. This discrepancy forces the network in the direction of $\Delta v \perp \nabla_x f_\theta$, leading to distortion of the flow map. The flow map induced by direct training is indefinite.*

This can affect methods that optimize the direct training objective. In such cases, injectivity and non-crossing trajectories are no longer guaranteed, which may result in mode collapse or failure of locality-based editing. Instead of optimizing the direct training objective, the consistency training objective ensures the marginal flow map at its fixed point, even when guided by conditional velocity. However, it does not guarantee convergence due to its gradient dynamics (proof in Appendix A.7).

**Proposition 3.4.** *(Instability of consistency training) The consistency training objective with a conditional velocity reduces to the objective with the marginal velocity under expectation.*

*However, it lacks the curvature required to stabilize the optimum, ensuring only the existence of a fixed point that satisfies the Eulerian equation rather than guaranteeing the global optimum. Thus, the gradient dynamics may fail to converge.*

Some work (Issenhuth et al., 2025; Silvestri et al., 2025) leverages neural networks to conduct the flow. In direct training settings, these approaches can provide a tighter bound to Eulerian distillation, but still do not guarantee the convergence to the marginal velocity field. (proof in Appendix A.8).

**Preconditioners.** Despite various training techniques proposed in prior work, classical consistency training still suffers from reproducibility issues. In particular, recent consistency training methods rely on initialization with pretrained diffusion models, such as Karras et al. (2022); Yao et al. (2025). These pretrained models are often referred to as multi-step *preconditioners*, and we observe that the performance of consistency models varies depending on them.

Table 2: **Consistency training results** on ImageNet $256 \times 256$ under different preconditioners. *Multi-step FID* denotes the FIDs of pretrained networks for given ODE solver and sampling-step pairs. *Few-step FID* denotes the 2-NFE FID of consistency models initialized from the corresponding preconditioner (details in Appendix C.1).

| Preconditioner | Multi-step FID↓ | Few-step FID↓ |
|---|---|---|
| Multi-step Baseline | 1.21 (UCGM-S, 30-step) | 2.69 |
| LightningDiT | 2.17 (Euler, 250-step) | 10.01 |
| Reproduced Model | 2.41 (UCGM-S, 30-step) | 5.96 |
| w/o Preconditioner | - | Diverged (200↑) |
| Reported Baseline | 1.21 (UCGM-S, 30-step) | 1.42 |

As shown in Tab. 2, we evaluated an open-source continuous-time consistency model (Sun et al., 2025) on ImageNet $256 \times 256$. With the released multi-step baseline, the model achieved a 2-NFE FID of 2.69. However, when using the pretrained LightningDiT (Yao et al., 2025) or our reproduced multi-step model, the FIDs were worse than the baseline and diverged when initialized randomly.

**Linearization Cost Hypothesis.** Some studies (Geng et al., 2025a; Frans et al., 2025) enable training from scratch without a preconditioner. The key difference is that they allow $s < t$, while others fix $s = 0$. Intuitively, training long-range mappings is more challenging than short-range ones, since the linearization cost increases with step size. In both objectives, we observe that $s \to t$ amplifies the flow matching term, while $s \to 0$ amplifies a linearization term involving JVP, which is structurally more complex (Appendix. A.9). We hypothesize that fixing $s = 0$ makes optimization more challenging, training less stable, while relaxing to $s < t$ balances the terms and mitigates instability.

## 4 TOWARDS REPRODUCIBLE AND STABLE FLOW MAP TRAINING

From these observations, we identify two key factors for reproducibility: relaxation of $s$ (linearization cost hypothesis), and marginal velocity guidance (Prop. 3.3). To facilitate consistency training from scratch, we relax the time condition to mitigate instability, and leverage self-distillation to follow the marginal velocity, addressing suboptimality. Since prior work on self-distillation (Boffi et al., 2025a) relies on heuristics to stabilize training, we propose *improved Self-Distillation (iSD)*: (i) reducing reliance on heuristics and simplifying the training process by exploring the design space of flow maps, (ii) extending classifier-free guidance to flow maps, achieving additional improvements.

### 4.1 FACILITATING CONSISTENCY TRAINING FROM SCRATCH

**Relaxation of $s$.** Based on the linearization cost hypothesis, we relax $s = 0$ to $s < t$, balancing the contributions of the flow matching and linearization terms. Instead of directly addressing the unstable gradient dynamics of consistency training, we leverage this relaxation to indirectly mitigate the instability. This approach still avoids the Hessian, while empirically stabilizing optimization.

**Marginal velocity guidance.** Following Prop. 3.3, we consider marginal velocity guidance to guarantee convergence. From the instantiations in Prop. 3.2, we identify that self-distillation follows the marginal velocity approximated by the network itself. Based on this, we train $F_\theta(x_t; t, t)$ via flow matching, while jointly applying Eulerian distillation to $F_\theta(x_t; t, s)$ guided by its approximation:

$$\mathcal{L}_{\text{CFM}} = \mathbb{E}\left[\|F_\theta(x_t; t, t) - v(x_t|x)\|_2^2\right] \tag{12}$$

$$\mathcal{L}_{\text{SD}} = \mathbb{E}\left[\|\partial_t f_\theta(x_t; t, s) + F_{\theta^-}(x_t; t, t) \cdot \nabla_x f_\theta(x_t; t, s)\|_2^2\right] \tag{13}$$

With this setting, the objective ensures convergence to the marginal flow map, handling suboptimality of direct training (proof in Appendix B.1; see also Boffi et al. (2025a)). Under consistency training, this can reduce loss variance and further stabilize training compared to the conditional velocity guidance (details in Appendix B.2).

### 4.2 IMPROVED SELF-DISTILLATION

To incorporate various design choices and reduce the reliance on heuristics, we extend the self-distillation method using our generalized formulations and the explored design space. In the next section, we conduct ablation studies across these choices and present the best practice.

**Reformulation.** We leverage our generalized flow map defined in Eq. 6, and the objective can be expressed using the guidance velocity $v_\theta(x_t; t) = F_\theta(x_t; t, t)$ as:

$$\mathcal{L}_{\text{SD}} = \mathbb{E}_{x,z,t,s}\left[\nu^{-2}\left\|A''_{t,s}x_t + A'_{t,s}(v_\theta(x_t; t) - F_\theta(x_t; t, s)) - A_{t,s}\frac{dF_\theta(x_t; t, s)}{dt}\right\|_2^2\right] \tag{14}$$

This extends the original self-distillation to arbitrary interpolations satisfying our assumptions. Next, we reformulate the objective, where the gradient remains identical while avoiding the Hessian:

$$\mathcal{L}_{\text{SD-R}} = \mathbb{E}_{x,z,t,s}\left[w_{t,s}\|F_\theta(x_t; t, s) - \text{sg}\left[F_{tgt}(x_t; t, s)\right]\|_2^2\right], \quad w_{t,s} = A_{t,s}\nu^{-2} \tag{15}$$

$$F_{\text{tgt}} = F_\theta(x_t; t, s) + A''_{t,s}x_t + A'_{t,s}(v_\theta(x_t; t) - F_\theta(x_t; t, s)) - A_{t,s}\frac{dF_\theta(x_t; t, s)}{dt} \tag{16}$$

**JVP Approximation.** Some works approximate the JVP, highlighting a trade-off between accuracy and computational efficiency. In our work, we approximate the JVP as follows to preserve the trajectory of marginal guidance. With this approximation, we found that training time can be reduced and performance further improves.

$$\frac{dF_\theta(x_t; t, s)}{dt} = \frac{F_\theta(x_t + \epsilon \cdot v_\theta(x_t; t); t + \epsilon, s) - F_\theta(x_t - \epsilon \cdot v_\theta(x_t; t); t - \epsilon, s)}{2\epsilon} \quad (17)$$

**Adaptive Weighting.** Since the JVP exhibits instability and can cause training to diverge, several ideas have been proposed to stabilize the operation (Lu & Song, 2025; Sun et al., 2025; Boffi et al., 2025a; Geng et al., 2025a). Among these, we adopt adaptive weighting to preserve the intended guidance, as normalization and clipping may alter the guiding trajectory. To ensure stable joint training, we extend weighting to both objectives and formulate it with hyperparameters $\eta$ and $p$.

$$\tilde{\mathcal{L}}_{t,s}(x_t, x) = \|F_\theta(x_t; t, t) - v_t(x_t|x)\|_2^2 + \|F_\theta(x_t; t, s) - \text{sg}[F_{\text{tgt}}(x_t; t, s)]\|_2^2$$
$$\mathcal{L}_{\text{iSD}} = \mathbb{E}_{x,z,t,s}\left[w_{t,s}(x_t, x) \cdot \tilde{\mathcal{L}}_{t,s}(x_t, x)\right], \quad w_{t,s}(x_t, x) = (\text{sg}[\tilde{\mathcal{L}}_{t,s}(x_t, x)] + \eta)^{-p} \quad (18)$$

From these settings, *training from scratch becomes stable, allowing us to eliminate additional heuristics introduced in prior work*, such as progressive distillation, annealing (Boffi et al., 2025a), small Fourier coefficients, double normalization, tangent warmup, and some regularizations (Sabour et al., 2025; Lu & Song, 2025; Chen et al., 2025). Detailed training and sampling algorithms are provided in appendix, Alg. 2 and Alg. 3.

**Classifier-free Guidance.** Classifier-free Guidance (CFG) is an off-the-shelf method for boosting the performance of diffusion models. However, unlike in diffusion models, directly applying CFG to flow maps does not guarantee mappings along the CFG velocity field (details in Appendix B.3).

This motivates two extensions of the proposed self-distillation: *Post-CFG* and *Pre-CFG*. Post-CFG operates as classical classifier-free guidance applied after training, defined as follows:

$$\tilde{F}_\theta(x_t; t, s, c) = F_\theta(x_t; t, s, \varnothing) + \omega(F_\theta(x_t; t, s, c) - F_\theta(x_t; t, s, \varnothing))$$
$$\tilde{f}_\theta(x_t; t, s, c) = \nu^{-1}(A'_{t,s}x_t - A_{t,s}\tilde{F}_\theta(x_t; t, s, c)) \quad (19)$$

where $\varnothing$ is the null class label for unconditional generation, $c$ is the conditional class label, and $\omega$ is the guidance scale. Although this formulation is not guaranteed to follow the CFG velocity field, it can be easily applied after training. To ensure that the flow map follows the CFG field, Pre-CFG replaces the guidance velocity $v_\theta$ with the CFG velocity $\tilde{v}_\theta$ during training:

$$\tilde{v}_\theta(x_t; t, c) = F_\theta(x_t; t, t, \varnothing) + \omega(F_\theta(x_t; t, t, c) - F_\theta(x_t; t, t, \varnothing)) \quad (20)$$

The CFG velocity is Lipschitz continuous under our assumption. Thus, our propositions also apply, guaranteeing convergence to the CFG velocity field.

However, since the ground-truth CFG velocity is intractable during training, applying $\mathcal{L}_{\text{CFM}}$ with the CFG velocity is infeasible. If we perform $\mathcal{L}_{\text{CFM}}$ with the conditional velocity, Pre-CFG causes a conflict: $\mathcal{L}_{\text{SD-R}}$ induces $F_{t,t} \approx \tilde{v}_t$, whereas $\mathcal{L}_{\text{CFM}}$ induces $F_{t,t} \approx v_t^*$. Therefore, we consider two cases: (i) $\mathcal{L}_{\text{iSD-U}}$, applying $\mathcal{L}_{\text{CFM}}$ with the conditional velocity while compromising the theoretical guarantees at $s = t$ (Guidance-**U**nconditional), and (ii) $\mathcal{L}_{\text{iSD-C}}$, appending the guidance scale as an additional condition $F_{t,t}^\theta(x_t; c, w)$, thereby applying $\mathcal{L}_{\text{CFM}}$ with $w = 1.0$ and $\mathcal{L}_{\text{SD-R}}$ with $w = \omega$ (Guidance-**C**onditional). In this case, $\mathcal{L}_{\text{iSD-C}}$ ensures $F_{t,t}(x_t; c, 1.0) \approx v_t^*(x_t; c)$ and $F_{t,t}(x_t; c, \omega) \approx \tilde{v}_t(x_t; c)$ (details in Appendix B.3). We discuss their practical benefits and present our final choice in the next section.

## 5 EXPERIMENTS

**Experimental Settings.** To evaluate our method, we conduct experiments on the ImageNet-1K (Deng et al., 2009) and CIFAR-10 (Krizhevsky, 2009) datasets. Following prior work, we use downsampled $32\times32\times4$ latent variables from $256\times256$ images encoded by a VAE (Rombach et al., 2022), and employ a DiT (Peebles & Xie, 2023) architecture. For CIFAR-10, we train the model directly in pixel space using UNet+ (Song et al., 2021). We evaluate both one-step and few-step generations using uniformly sampled timesteps. Sample quality is measured with FID (Heusel et al., 2017) over 50K samples, and further implementation details are provided in Appendix C.2.

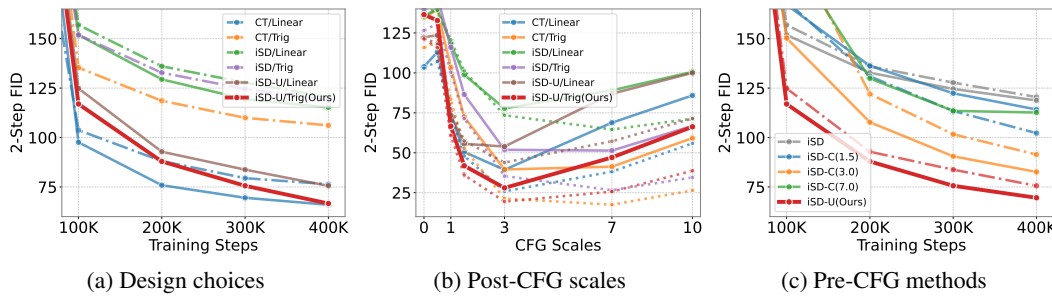

Figure 2: **Design choices of the generalized flow map.** (a) FIDs of design choices over training steps. Solid lines indicate the JVP approximation, and dash-dot lines indicate direct JVP. (b) FIDs of Post-CFG over guidance scales. Dotted lines indicate 4-Step FIDs. (c) FIDs of Pre-CFG over training steps. Solid lines indicate trigonometric interpolation and dash-dot lines indicate linear one.

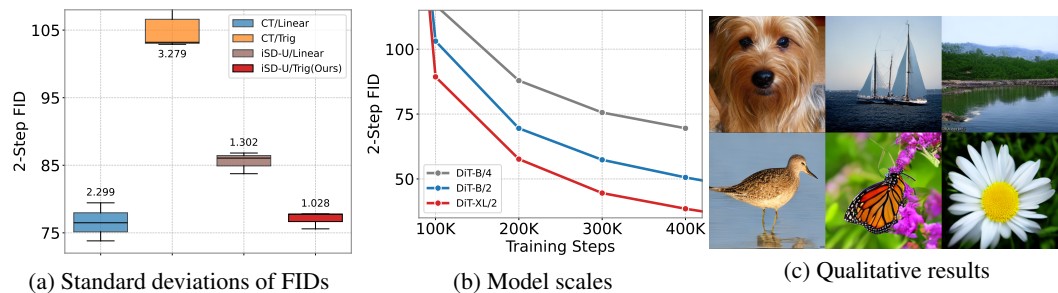

Figure 3: **Few-step generation results**. (a) Standard deviations of FIDs at 300K training steps. (b) FIDs of model scales over training steps. (c) 4-Step samples generated by iSD-U.

Table 3: **Step-by-step experiments validating the hypothesis** (2-Step).

| Case | FID↓ |
|---|---|
| Baseline | 121.3 |
| + Joint training $\mathcal{L}_{\text{CFM}}$ | 116.3 |
| + Relaxing condition $\mathcal{L}_{\text{CT}}$ | 76.39 |
| + Self-Distillation ($\omega = 1.5$) | 75.57 |
| + Trigonometric interpolation | 66.63 |
| Standard deviation | 0.735 |

Table 4: **Quantitative results across design choices** (2-Step). The numeric entries in the header denote Post-CFG scales.

| Loss | Interp. | JVP | FID↓ | 1.5 | 3.0 | 7.0 | 10.0 |
|---|---|---|---|---|---|---|---|
| $\mathcal{L}_{\text{CT}}$ | Linear | Exact | 76.39 | 50.45 | 39.33 | 68.82 | 85.87 |
| $\mathcal{L}_{\text{CT}}$ | Trig. | Exact | 103.35 | 72.42 | 39.52 | 41.27 | 59.16 |
| $\mathcal{L}_{\text{iSD}}$ | Linear | Exact | 118.17 | 98.77 | 77.75 | 89.02 | 100.61 |
| $\mathcal{L}_{\text{iSD}}$ | Trig. | Exact | 115.93 | 86.46 | 51.85 | 51.34 | 66.79 |
| $\mathcal{L}_{\text{CT}}$ | Linear | Approx | 65.98 | 40.97 | 33.33 | 67.99 | 85.69 |
| $\mathcal{L}_{\text{iSD}}$ | Linear | Approx | 112.39 | 90.26 | 69.81 | 87.50 | 101.42 |
| $\mathcal{L}_{\text{iSD-U}}$ | Linear | Approx | 75.57 | 55.58 | 53.88 | 86.95 | 99.97 |
| $\mathcal{L}_{\text{iSD-U}}$ | Trig. | Approx | 66.63 | 41.84 | **27.99** | 47.04 | 66.25 |

## 5.1 ABLATION STUDY

We conduct our ablation study on DiT-B/4, a base model of diffusion transformer with $4 \times 4$ patches. The model is trained for 400K steps with a batch size of 256. By default, we set $\omega = 1.5$ for $\mathcal{L}_{\text{iSD-U}}$ and use conditional velocity guidance for $\mathcal{L}_{\text{CT}}$.

**Key factors.** To validate our hypothesis, we conduct step-by-step experiments to make consistency training reproducible with our contributions, as summarized in Tab. 3. Starting from the consistency-training baseline with a linear trajectory, joint training with flow matching improves the FID. Relaxing the time condition further reduces it, supporting the linearization cost hypothesis. We observe that self-distillation converges more slowly than consistency training, as shown in Tab. 4 and Fig. 2a. Pre-CFG $\mathcal{L}_{\text{iSD-U}}$ accelerates training compared to vanilla $\mathcal{L}_{\text{iSD}}$, and achieves further improvement when the linear interpolation is replaced with a trigonometric one.

**Interpolation, Post-CFG.** As shown in Fig. 2a and Tab. 4, under conditional velocity guidance, linear interpolation yields better results compared to the trigonometric case. However, with self-distillation, the trigonometric interpolation achieves a lower FID than linear. It exhibits a larger performance gap under the Post-CFG (Fig. 2b), even surpassing the linear case at 4-step sampling. Pre-CFG $\mathcal{L}_{\text{iSD-U}}$ also achieves better results with the trigonometric case.

**JVP operation.** When comparing the JVP approximation with direct computation, the approximation achieves better results. All subsequent experiments adopt the JVP approximation by default.

Table 5: **Comparison with prior work** on ImageNet $256 \times 256$ and CIFAR-10.

| Class-Conditional ImageNet $256 \times 256$ | | |
|---|---|---|
| **METHOD** | **NFE** ($\downarrow$) | **FID** ($\downarrow$) |
| **Consistency Training Methods** | | |
| iCT (Song & Dhariwal, 2024) | 2 | 20.3 |
| Shortcut Model (Frans et al., 2025) | 1 | 10.6 |
| | 4 | 7.8 |
| UCGM (SD-VAE, Sun et al. (2025)) | 1 | 2.10 |
| MeanFlow (Geng et al., 2025a) | 1 | 3.43 |
| **Ours** | | |
| iSD-U | 2 | 23.51 |
| | 4 | 20.43 |
| + Post-CFG ($\omega = 3.0$) | $2 \times 2$ | 13.49 |
| | $4 \times 2$ | 11.06 |

| Unconditional CIFAR-10 | | |
|---|---|---|
| **METHOD** | **NFE** ($\downarrow$) | **FID** ($\downarrow$) |
| **Distillation-based Methods** | | |
| 2-RF Liu et al. (2023) | 1 | 4.85 |
| DMD (Yin et al., 2024b) | 1 | 3.77 |
| **Consistency Training Methods** | | |
| iCT (Song & Dhariwal, 2024) | 1 | 2.83 |
| sCT (Lu & Song, 2025) | 1 | 2.97 |
| UCGM (Sun et al., 2025) | 1 | 2.82 |
| MeanFlow (Geng et al., 2025a) | 1 | 2.92 |
| Self-distillation (Boffi et al., 2025a) | 1 | 14.13 |
| **Ours** | | |
| iSD | 1 | 3.64 |

**Pre-CFG.** As shown in Fig. 2c and Tab. 4, applying Pre-CFG $\mathcal{L}_{\text{iSD-U}}$ with $\omega = 1.5$ yields improved FIDs compared to vanilla $\mathcal{L}_{\text{iSD}}$, but training diverged when $\omega > 3.0$. When a guidance scale is appended as a condition, $\mathcal{L}_{\text{iSD-C}}$ enables training at higher guidance scales, outperforming $\mathcal{L}_{\text{iSD}}$. However, $\mathcal{L}_{\text{iSD-C}}$ consistently performs worse results than $\mathcal{L}_{\text{iSD-U}}$. Intuitively, the additional condition enforces the network to learn both CFG and non-CFG mappings, imposing an extra burden on the network. Even though $\mathcal{L}_{\text{iSD-U}}$ compromises the theoretical guarantees at $s = t$, few-step generation commonly assumes $s \ll t$, making this negligible in practice. Thus, we finalize the design choices: (i) JVP approximation, (ii) trigonometric interpolation, and (iii) Pre-CFG $\mathcal{L}_{\text{iSD-U}}$.

**Reproducibility** To validate the reproducibility of our method, we measure the variance of FIDs across three runs with different random initializations. We compare our final version of iSD with consistency training baseline, involving direct JVP and $s < t$. As shown in Fig. 3a, where the numbers in the box plot denote standard deviations, our method demonstrates improved reproducibility compared to the consistency training baseline, while remaining competitive FIDs.

**Scalability.** Fig. 3b presents the results across model scales. As the patch size decreases from DiT-B/4 to DiT-B/2 and computation increases, the FID improves from 66.63 to 50.58. Further scaling from DiT-B/2 to DiT-XL/2 improves the FID to 38.50, demonstrating consistent scaling behavior.

## 5.2 COMPARISON WITH PRIOR WORK

In Tab. 5, we compare our work with previous methods on ImageNet $256 \times 256$. We train DiT-XL/2 with iSD-U for 800K steps following prior work. Our model demonstrates comparable result to our reproduced consistency model (Tab. 2, FID 10.01), iCT, and Shortcut Model, but higher FIDs than others. On CIFAR-10, we obtain improved results compared to the original self-distillation, reducing the FID from 14.13 to 3.64, while achieving performance comparable to other prior work.

Since our method requires neither additional networks nor pretrained models, the training process is simplified, and its reproducibility has been validated in the previous section. However, we found that training is slower than consistency training methods, since it first learns the marginal instantaneous velocity and then the flow map guided by itself. The training was not saturated even after 800K steps, and we leave further training for future work. Accelerating training also remains a promising direction.

## 6 CONCLUSION

We introduced a generalized flow map framework that unifies recent consistency-based generative models under the Eulerian equation. This highlights the suboptimality of existing approaches and explains their limited reproducibility. To address these issues, we propose *improved Self-Distillation*, which reduces reliance on heuristics and simplifies the training process. We further extend it to classifier-free guidance for flow maps, achieving additional performance gains. Empirically, our method achieves reproducible training and competitive few-step generation on ImageNet-1K. These results establish a theoretical and empirical foundation for reproducible consistency training.

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

APPENDIX

# A  THEORETICAL ANALYSIS OF FLOW MAP MODELS

## A.1  MEAN COLLAPSE OF DIFFUSION AND FLOW MATCHING MODELS

**Posterior Distribution.** First, consider the data distribution $p_X(x) = \mathcal{N}(\mu_X, \sigma_X^2)$ and the interpolation $x_t = \alpha_t x + \sigma_t z$. The conditional distribution is given by $p(x_t = y|x) = \mathcal{N}(y; \alpha_t x, \sigma_t^2)$.

By Bayes' rule,

$$
\begin{aligned}
p(x|x_t = y) &\propto p(x_t = y|x)p(x) \\
&= \mathcal{N}(y; \alpha_t x, \sigma_t^2)\mathcal{N}(x; \mu_X, \sigma_X^2) \\
&\propto \exp\left(-\frac{(y - \alpha_t x)^2}{2\sigma_t^2} - \frac{(x - \mu_X)^2}{2\sigma_X^2}\right) \\
&= \exp\left(\left(\frac{1}{2\sigma_X^2} + \frac{\alpha_t^2}{2\sigma_t^2}\right)x^2 - \left(\frac{\mu_X}{\sigma_X^2} + \frac{\alpha_t y}{\sigma_t^2}\right)x + \left(\frac{\mu^2}{2\sigma_X^2} + \frac{y^2}{2\sigma_t^2}\right)\right)
\end{aligned}
$$

This can be organized as a Gaussian with a closed form $p(x|x_t = y) = \mathcal{N}(\mu_{x|y,t}, \sigma_{x|y,t}^2)$ where

$$
\mu_{x|y,t} = \frac{\alpha_t \sigma_X^2 y + \mu_X \sigma_t^2}{\sigma_t^2 + \sigma_X^2 \alpha_t^2}, \quad \sigma_{x|s,t}^2 = \frac{\sigma_X^2 \sigma_t^2}{\sigma_t^2 + \sigma_X^2 \alpha_t^2}
$$

Extending the data distribution to a mixture of Gaussians $p_X(x) = \sum_i \pi_i \mathcal{N}(x; \mu_i, \sigma_i^2)$, we introduce the latent variable $\pi$ for handling $\pi_i$:

$$
p(\pi = i) = \pi_i, \quad p(x|\pi = i) = \mathcal{N}(x; \mu_i, \sigma_i^2)
$$

Then, the marginal distribution $p(x_t = y|\pi = i)$ can be expressed as

$$
\begin{aligned}
p(x_t = y|\pi = i) &= \int p(x_t = y|x)p(x|\pi = i)dx \\
&= \int \mathcal{N}(y; \alpha_t x, \sigma_t^2)\mathcal{N}(x; \mu_i, \sigma_i^2)dx \\
&= \mathcal{N}(y; \alpha_t \mu_i, \alpha_t^2 \sigma_i^2 + \sigma_t^2)
\end{aligned}
$$

And we define responsibilities $r_i(y)$ as posterior distribution

$$
r_{i,t}(y) = p(\pi = i|x_t = y) = \frac{p(x_t = y|\pi = i)p(\pi = i)}{\sum_j p(x_t = y|\pi = j)p(\pi = j)} = \frac{\pi_i \mathcal{N}(y; \alpha_t \mu_i, \alpha_t^2 \sigma_i^2 + \sigma_t^2)}{\sum_j \pi_j \mathcal{N}(y; \alpha_t \mu_j, \alpha_t^2 \sigma_j^2 + \sigma_t^2)}
$$

Therefore, the posterior distribution $p(x|x_t = y)$ is

$$
\begin{aligned}
p(x|x_t = y) &= \sum_i p(\pi = i|x_t = y)p(x|x_t = y, \pi = i) \\
&= \sum_i r_{i,t}(y)\mathcal{N}(y; \alpha_t x, \sigma_t^2)\mathcal{N}(x; \mu_i, \sigma_i^2) \\
&= \sum_i r_{i,t}(y)\mathcal{N}(x; \mu_{x|i,y,t}, \sigma_{x|i,y,t}^2)
\end{aligned}
$$

$$
\text{where } \mu_{x|i,y,t} = \frac{\alpha_t \sigma_i^2 y + \mu_i \sigma_t^2}{\sigma_t^2 + \sigma_i^2 \alpha_t^2}, \quad \sigma_{x|i,y,t}^2 = \frac{\sigma_i^2 \sigma_t^2}{\sigma_t^2 + \sigma_i^2 \alpha_t^2}
$$

Particularly, we observe that $\mu_{x|i,y,1} = \mu_i$, $\sigma_{x|i,y,1}^2 = \sigma_i^2$ and $r_{i,1}(y) = \pi_i$.

**One-step Generation.** Under the linear trajectory $x_t = (1 - t)x + tz$, the conditional velocity is $v_t(x_t|x) = z - x$. Thus, one-step generation is defined by

$$
\begin{aligned}
f_F(x_t; t) &= x_t - tF^*(x_t; t) = x_t - t\,\mathbb{E}_{x|x_t}[v_t(x_t|x)] \\
&= \mathbb{E}_{x,z|x_t}[x_t - t(z - x)] \\
&= \mathbb{E}_{x|x_t}[x]
\end{aligned}
$$

In the unimodal Gaussian case, $\mathbb{E}_{x|z}[x] = \mu_{x|z,1} = \mu_X$ and the one-step generated samples collapse to the mean of the data distribution. Similarly, in the mixture of Gaussians case, one-step generated samples collapse to the mixture mean.

$$f_F(z) = \mathbb{E}_{x|z}[x] = \sum_i r_i(z)\,\mu_{i|z,1} = \sum_i \pi_i\,\mu_i = \mu_X$$

Thus, one-step generation collapses to the data mean $\mu_X$ regardless of the input. $\qquad\square$

### A.2 Injectivity of Flow Map

Since the marginal velocity is assumed to be Lipschitz continuous, the Picard-Lindelöf theorem guarantees a unique solution to the ODE $dx_t = v_t^*(x_t)dt$ for any initial value. Non-crossing trajectory follows directly, since any crossing would contradict the uniqueness. Thus, since the flow map is formulated as the solution of the ODE with the initial value $x_t$, it is well-defined and the solution $x_s$ is uniquely determined by non-crossing, ensuring the injectivity of the flow map.

### A.3 Eulerian Equation and Uniqueness of Flow Map

Suppose the ground-truth flow map is defined as

$$f_{t,s}^*(x_t) = x_t + \int_t^s v_\tau^*(x_\tau)d\tau = x_s$$

By construction, the identity mapping $f_{t,s}^*(f_{s,t}^*(x_s)) = x_s$ satisfies. Differentiating both sides w.r.t. $t$ yields

$$\frac{d}{dt} f_{t,s}^*(f_{s,t}^*(x_s)) = \partial_t f_{t,s}^*(f_{s,t}^*(x_s)) + \partial_t f_{s,t}^*(x_s) \cdot \nabla_x f_{t,s}^*(f_{s,t}^*(x_s)) = \frac{d}{dt}x_s = 0$$

Using $f_{s,t}^*(x_s) = x_t$ and $\partial_t f_{s,t}^*(x_s) = \partial_t x_t = v_t^*(x_t)$, we obtain the Eulerian equation:

$$\frac{d}{dt} f_{t,s}^*(x_t) = \partial_t f_{t,s}^*(x_t) + v_t^*(x_t) \cdot \nabla_x f_{t,s}^*(x_t) = 0$$

Suppose a trainable network $f_\theta(x; t, s) = f_{t,s}^\theta(x)$ is continuous in $x, t, s$, Lipschitz continuous in $x$, and satisfies the boundary condition $f_{s,s}^\theta(x) = x$ for all $s$. If $f_\theta$ satisfies the Eulerian equation, $f_{t,s}^\theta$ remains constant along the characteristic curve induced by $v_t^*(x_t)$.

Let $\chi_\tau$ denotes the characteristic curve defined on $[s,t]$ by $\chi_t = x$ and $\chi_\tau' = v_\tau^*(\chi_\tau)$. Along this curve, $f_{\tau,s}^\theta$ is constant and evaluating at $\tau = t$ and $\tau = s$ yields

$$f_{t,s}^\theta(x) = f_{t,s}^\theta(\chi_t) = f_{s,s}^\theta(\chi_s) = \chi_s = f_{t,s}^*(x)$$

since $f_{t,s}^*$ generates the characteristic curve $\chi_\tau$ by its definition.

Thus, the learned mapping coincides with the exact flow map. $\qquad\square$

### A.4 Interpolation condition for guaranteeing the convergence

We begin by explicitly deriving the solution of the Eulerian equation. For $f_{t,s}^\theta(x_t) = \nu_t^{-1}(A_{t,s}' x_t - A_{t,s}F_{t,s}^\theta(x_t))$ in Eq. 6, let $n = A_{t,s}' x_t - A_{t,s}F_{t,s}^\theta(x_t)$, which simplifies to $n = A'x_t - AF$ and $f = \nu_t^{-1}n$. Then, differentiation with respect to timestep $t$ yields

$$n' = A''x_t + A'(v_t - F) - AF'$$

$$f' = [A''x_t + A'(v_t - F) - AF']\frac{\nu_t}{\nu_t^2} + [A'x_t - AF]\frac{\nu_t'}{\nu_t^2}$$

Assuming $f' = 0$ and $\nu_t \neq 0$, we obtain

$$A\nu_t F' = -(A'\nu_t + A\nu_t')F + [A''\nu_t x_t + A'\nu_t v_t + A'\nu_t' x_t]$$

which further simplifies to

$$A\nu_t F' = -[A\nu_t]'F + [A\nu_t x_t]' = BF' = -B'F + D'$$

where $B = A\nu_t$, $D = A'\nu_t x_t$. Since $BF' + B'F = [BF]'$, we obtain $[BF]' = D'$, which yields $A\nu_t F = A'\nu_t x_t + C$ for some integration constant $C$. Assuming $A = A_{t,s} = \alpha_s \sigma_t - \sigma_s \alpha_t \neq 0$ for $t \neq s$, this gives $F = A'A^{-1}x_t + C(A\nu_t)^{-1}$, and hence

$$f_{t,s}^*(x_t) = \nu_t^{-1}[A'x_t - A'x_t - C\nu_t^{-1}] = -C\nu_t^{-2}$$

If $\nu_t$ is a time-dependent scalar, the global optimum $f^* = -C'\nu_t^{-2}$ is itself time-dependent, so the Eulerian equation can't vanish. If $C = 0$, the solution collapses to the trivial case $f_{t,s}^*(x_t) = 0$. Therefore, $\nu_t$ must be a time-independent constant. $\qquad\square$

**Additional Observation.** Suppose that the monotonically increasing $\gamma_t$ over $t \in [0,1]$ satisfying the boundary conditions $\gamma_1 = 1$ and $\gamma_0 = 0$. Consider the interpolation defined by $\alpha_t = (1 - \gamma_t)^c$ and $\sigma_t = \gamma_t^c$ for some constant $c \in [0.5, 1]$. Then, $\nu_t$ can be written as $\nu_t = c(1 - \gamma_t)^{c-1}\gamma_t^{c-1}\gamma_t'$. Imposing $\nu_t = \nu$ gives $\gamma_t' = \nu[c(1 - \gamma_t)^{c-1}\gamma_t^{c-1}]^{-1}$ and $c(1 - \gamma)^{c-1}\gamma^{c-1}d\gamma = \nu dt$. Integrating both sides yields

$$c\int (1 - \gamma)^{c-1}\gamma^{c-1}d\gamma = \nu \int dt = \nu(t + C)$$

where the constant $C$ vanishes due to $\gamma_0 = 0$. For the incomplete beta function $B$, this becomes

$$cB(\gamma_t; c, c) = c\int_0^{\gamma_t} (1 - \eta)^{c-1}\eta^{c-1}d\eta = \nu \int_0^t d\tau = \nu t$$

By the boundary condition, $cB(c, c) = cB(1; c, c) = \nu$, and thus $\gamma_t$ is characterized by

$$cB(\gamma_t; c, c) = \nu\frac{cB(\gamma_t; c, c)}{cB(c, c)} = \nu I_{\gamma_t}(c, c) = \nu t \implies \gamma_t = I_t^{-1}(c, c)$$

where $I$ denotes the regularized incomplete beta function.

In this case, $\gamma_t$ is characterized regardless of $\nu$. Particularly, when $c = 0.5$, we obtain $\gamma_t = \sin^2(\frac{\pi}{2}t)$, which yields trigonometric interpolation $\alpha_t = \cos(\frac{\pi}{2}t)$ and $\sigma_t = \sin(\frac{\pi}{2}t)$. On the other hand, when $c = 1$, we have $\gamma_t = t$, $\alpha_t = 1 - t$, $\sigma_t = t$, and which reduces to linear interpolation. Interpolating $c$ between 0.5 and 1.0 provides a promising design choice for formulating the consistency model.

## A.5 RECENT CONSISTENCY-BASED GENERATIVE MODELS ARE FLOW MAP MODELS

**sCT.** Under trigonometric interpolation $x_t = \cos(t)x + \sin(t)z$,

$$f_\theta(x_t; t, s) = \cos(s - t)x_t + \sin(s - t)F_\theta(x_t; t, s)$$

When $s = 0$

$$f_\theta(x_t; t) = \cos(t)x_t - \sin(t)F_\theta(x_t; t)$$

which exactly recovers the sCT formulation. If consistency training is formulated without the stop-gradient operation, then the objective reduces to the direct training objective as $\Delta t \to 0$.

$$\mathbb{E}\left[\|f_\theta(x_t; t, s) - f_\theta(x_{t-\Delta t}; t - \Delta t, s)\|_2^2\right]$$

$$= \mathbb{E}\left[\|f_\theta(x_t; t, s) - [f_\theta(x_t; t, s) - \partial_t f_\theta(x_t; t, s) \cdot \Delta t - \nabla_x f_\theta(x_t; t, s) \cdot v(x_t|x) \cdot \Delta t + O(\Delta t^2)]\|_2^2\right]$$

$$= \Delta t^2 \cdot \mathbb{E}\left[\|\partial_t f_\theta(x_t; t, s) + v(x_t|x) \cdot \nabla_x f_\theta(x_t; t, s)\|_2^2\right] + O(\Delta t^3)$$

However, if we utilize the stop-gradient, the continuous-time consistency training objective is defined as

$$\nabla_\theta \mathbb{E}\left[\|f_\theta(x_t; t, s) - f_{\theta^-}(x_{t-\Delta t}; t - \Delta t, s)\|_2^2\right]$$

$$= \mathbb{E}\left[2\nabla_\theta f_\theta(x_t; t, s) \cdot (f_{\theta^-}(x_t; t, s) - f_{\theta^-}(x_{t-\Delta t}; t - \Delta t, s))\right]$$

$$= 2\Delta t \cdot \nabla_\theta \mathbb{E}\left[f_\theta(x_t; t, s) \cdot \frac{f_{\theta^-}(x_t; t, s) - f_{\theta^-}(x_{t-\Delta t}; t - \Delta t, s)}{\Delta t}\right]$$

$$\implies \mathcal{L}_{\text{CT}} = \mathbb{E}\left[f_\theta(x_t; t, s)\frac{df_{\theta^-}(x_t; t, s)}{dt}\right]$$

By the mean value theorem,

$$f_{\theta^-}(x_t; t, s) = f_\theta(x_t; t, s) + \nabla_\theta f_\zeta(x_t; t, s) \cdot (\theta^- - \theta)$$

for the parameter $\zeta$ lying between $\theta$ and $\theta^-$. Using this, we can show that

$$\mathbb{E}\left[\left\|f_\theta(x_t; t, s) - f_{\theta^-}(x_{\hat{t}}; \hat{t}, s)\right\|_2^2\right]$$

$$= \mathbb{E}\left[\left\|f_\theta(x_t; t, s) - f_\theta(x_{\hat{t}}; \hat{t}, s) - \nabla_\theta f_\zeta(x_{\hat{t}}; \hat{t}, s) \cdot (\theta^- - \theta)\right\|_2^2\right]$$

$$= \mathbb{E}\left[\left\|\partial_t f_\theta(x_t; t, s) \cdot \Delta t + \nabla_x f_\theta(x_t; t, s) \cdot v(x_t|x) \cdot \Delta t - \nabla_\theta f_\zeta(x_{\hat{t}}; \hat{t}, s) \cdot (\theta^- - \theta) + O(\Delta t^2)\right\|_2^2\right]$$

$$= \mathcal{L}_{\text{DT}} \cdot \Delta t^2 - 2\mathbb{E}\left[Lf_\theta(x_t; t, s) \cdot J_{\hat{t}, s}^\zeta \cdot \Delta\theta\right] \cdot \Delta t + \mathbb{E}\left[\left\|J_{\hat{t}, s}^\zeta \cdot \Delta\theta\right\|_2^2\right] + O(\Delta t^3) + O(\Delta t^2 \|\Delta\theta\|)$$

where $\hat{t} = t - \Delta t$, $\Delta\theta = \theta^- - \theta$, $J_{\hat{t}, s}^\zeta = \nabla_\theta f_\zeta(x_{\hat{t}}; \hat{t}, s)$, and the operator is defined as $Lf = \partial_t f + v \cdot \nabla_x f$. In this case, since $f$ is Lipschitz and has a bounded first derivative, $Lf$ and $J$ are bounded. Hence, $2(Lf \cdot \Delta t - J \cdot \Delta\theta) \cdot O(\Delta t^2) = O(\Delta t^3) + O(\Delta t^2 \|\Delta\theta\|)$

If we set $\theta^- = \text{sg}[\theta]$, this reduces to

$$\mathbb{E}\left[\|f_\theta(x_t; t, s) - f_{\theta^-}(x_{t-\Delta t}; t - \Delta t, s)\|_2^2\right] = \mathcal{L}_{\text{DT}} \cdot \Delta t^2 + O(\Delta t^3)$$

Thus, the formulation can be interpreted as the direct training objective.

From another perspective, since the time derivative of $f_{t,s}^\theta(x_t) = f_\theta(x_t; t, s)$ is given by

$$\frac{df_{t,s}^\theta(x_t)}{dt} = \partial_t f_{t,s}^\theta(x_t) + v(x_t|x) \cdot \nabla_x f_{t,s}^\theta(x_t) = (L_* f_{t,s}^\theta)(x_t) + \Delta v \cdot \nabla_x f_{t,s}^\theta(x_t)$$

where $\Delta v = v_t(x_t|x) - v_t^*(x_t)$ and $L_* f_{t.s} = \partial_t f_{t,s} + v_t^* \cdot \nabla_x f_{t,s}$, the objective can be written as

$$\mathcal{L}_{\text{CT}} = \mathbb{E}\left[f_{t,s}^\theta\left(L_* f_{t,s}^{\theta^-}\right) + f_{t,s}^\theta\left(\Delta v \cdot \nabla_x f_{t,s}^{\theta^-}\right)\right]$$

The first term on the right-hand side corresponds to Eulerian distillation. By the tower property, the second term vanishes under conditional expectation:

$$\mathbb{E}_{x,z,t,s}\left[f_{t,s}^\theta(\Delta v \cdot \nabla_x f_{t,s}^{\theta^-})\right] = \mathbb{E}_{x,z,t,s}\left[\mathbb{E}_{\tilde{x}|x_t}\left[f_{t,s}^\theta(\Delta v \cdot \nabla_x f_{t,s}^{\theta^-})\right]\right]$$

$$= \mathbb{E}_{x,z,t,s}\left[f_{t,s}^\theta\left(\mathbb{E}_{\tilde{x}|x_t}[\Delta v] \cdot \nabla_x f_{t,s}^{\theta^-}\right)\right]$$

$$= \mathbb{E}_{x,z,t,s}\left[f_{t,s}^\theta \cdot 0 \cdot \nabla_x f_{t,s}^{\theta^-}\right] = 0$$

Thus, $\mathcal{L}_{\text{CT}}$ reduces in value to Eulerian distillation even along the conditional trajectory. However, the gradients of Eulerian distillation and consistency training differ, and their training dynamics may therefore exhibit distinct behaviors. The instability of these gradient dynamics is discussed in Appendix A.7. □

**MeanFlow.** Suppose a flow map model under linear interpolation.

$$f_\theta(x_t; t, s) = x_t + (s - t)F_\theta(x_t; t, s), \ x_t = (1 - t)x + tz$$

The corresponding direct training objective is

$$\mathcal{L}(\theta) = \mathbb{E}\left[\left\|\partial_t f_{t,s}^\theta(x_t) + v(x_t|x) \cdot \nabla_x f_{t,s}^\theta(x_t)\right\|_2^2\right] = \mathbb{E}\left[\left\|\frac{df_{t,s}^\theta(x_t)}{dt}\right\|_2^2\right]$$

where

$$\frac{d}{dt}f_\theta(x_t; t, s) = v_t - F_\theta(x_t; t, s) + (s - t)\frac{d}{dt}F_\theta(x_t; t, s)$$

Recall the MeanFlow objective from Geng et al. (2025a)

$$\mathcal{L}(\theta) = \mathbb{E}[\|u_\theta(z_t; r, t) - \text{sg}[v_t - (t - r)(v_t \cdot \partial_z u_\theta + \partial_t u_\theta)]\|_2^2]$$

Rewrite the MeanFlow objective by using flow map notation and transform

$$\nabla_\theta \mathbb{E}\left[\|F_\theta(x_t; t, s) - \mathrm{sg}[v_t(x_t|x) - (t - s)(v_t(x_t|x) \cdot \partial_z F_\theta + \partial_t F_\theta)]\|_2^2\right]$$

$$= \nabla_\theta \mathbb{E}\left[\left\|v_t(x_t|x) - F_\theta(x_t; t, s) + (s - t) \cdot \frac{d}{dt} F_{\theta^-}(x_t; t, s)\right\|_2^2\right]$$

$$= \nabla_\theta \mathbb{E}\left[\left\|F_\theta(x_t; t, s) - F_{\theta^-}(x_t; t, s) - \left[v_t(x_t|x) - F_{\theta^-}(x_t; t, s) + (s - t)\frac{d}{dt} F_{\theta^-}(x_t; t, s)\right]\right\|_2^2\right]$$

$$= \nabla_\theta \mathbb{E}\left[\frac{1}{t - s} f_\theta(x_t; t, s) \frac{df_{\theta^-}(x_t; t, s)}{dt}\right]$$

Thus, the MeanFlow objective is a special case of the continuous-time consistency training with conditional velocity under linear interpolation. □

**Shortcut Model.** From Frans et al. (2025), the Shortcut Model objective consists of the flow matching objective and the consistency objective.

$$\mathcal{L}(\theta) = \mathbb{E}[\|s_\theta(x_t; t, 0) - v_t\|_2^2 + \|s_\theta(x_t; t, 2d) - [s_\theta(x_t; t, d) + s_\theta(x'_{t+d}; t + d, d)]/2\|_2^2]$$

with $x'_{t+d} = x_t + s_\theta(x_t, t, d)$. By setting $d = s - t$ and $F_\theta(x_t; t, s) = s_\theta(x_t; t, s - t)$, sampling $t \sim \mathcal{U}[0, 1]$, and choosing $s = t - 2^{-d'}$ for $d' \sim Cat[1, 7]$, we obtain the flow map under linear interpolation

$$f_\theta(x_t; t, s) = x_t + (s - t)F_\theta(x_t; t, s) = x'_{t+d}$$

We can rewrite the flow matching objective of the Shortcut model as

$$\|s_\theta(x_t; t, 0) - v_t\|_2^2 = \|F_\theta(x_t; t, t) - v_t(x_t|x)\|_2^2$$

For $r = s + d$, the consistency objective of the Shortcut Model can be written in the form of the CTM(Consistency Trajectory Model)

$$\|s_\theta(x_t; t, 2d) - [s_\theta(x_t; t, d) + s_\theta(x'_{t+d}; t + d, d)]/2\|_2^2$$

$$= \|F_\theta(x_t; t, r) - [F_\theta(x_t; t, s) + F_\theta(f_\theta(x_t; t, s); s, r)]/2\|_2^2$$

$$= \frac{1}{4d^2}\|x_t + 2d \cdot F_\theta(x_t; t, r) - x_t - 2d[F_\theta(x_t; t, s) + F_\theta(f_\theta(x_t; t, s); s, r)]/2\|_2^2$$

$$= \frac{1}{4d^2}\|x_t + 2d \cdot F_\theta(x_t; t, r) - [x_t + d \cdot F_\theta(x_t; t, s) + d \cdot F_\theta(f_\theta(x_t; t, s); s, r)]\|_2^2$$

$$= \frac{1}{4d^2}\|f_\theta(x_t; t, r) - [f_\theta(x_t; t, s) + d \cdot F_\theta(f_\theta(x_t; t, s); s, r)]\|_2^2$$

$$= \frac{1}{4d^2}\|f_\theta(x_t; t, r) - f_\theta(f_\theta(x_t; t, s); s, r)\|_2^2$$

Hence, the objective of the Shortcut Model is

$$\mathcal{L}(\theta) = \mathbb{E}\left[\|F_\theta(x_t; t, t) - v_t\|_2^2 + \frac{1}{4d^2}\|f_\theta(x_t; t, r) - f_\theta(f_\theta(x_t; t, s); s, r)\|_2^2\right]$$

With the Taylor approximation of $F_{t,s} = F_{t,s}(x_t) = F_\theta(x_t; t, s)$

$$F_{t,r} = F_{t,s} + d \cdot \partial_s F_{t,s} + O(d^2), \quad F_{s,r} = F_{t,s} + d \cdot \partial_t F_{t,s} + d \cdot \partial_s F_{t,s} + d \cdot F_{t,s}^T \nabla_x F_{t,s} + O(d^2)$$

we obtain

$$d[2F_{t,r} - F_{t,s} - F_{s,r}]$$

$$= d[2[F_{t,s} + d \cdot \partial_s F_{t,s}] - F_{t,s} - [F_{t,s} + d \cdot \partial_t F_{t,s} + d \cdot \partial_s F_{t,s} + d \cdot F_{t,s}^T \nabla_x F_{t,s}]] + O(d^3)$$

$$= d^2[\partial_s F_{t,s} - \partial_t F_{t,s} - F_{t,s}^T \nabla_x F_{t,s}] + O(d^3)$$

Thus,

$$\frac{1}{4d^2}\|f_{t,r}(x_t) - f_{s,r}(f_{t,s}(x_t))\|_2^2 = \frac{d^2}{4}\|\partial_s F_{t,s} - \partial_t F_{t,s} - F_{t,s}^T \nabla_x F_{t,s}\|_2^2 + O(d^3)$$

The differentiation of the linear flow map with respect to timestep $t$ is given by

$$f'_{t,s}(x_t) = v_t^* - F_{t,s}(x_t) + (s - t) \cdot (\partial_t F_{t,s} + v_t^* \cdot \nabla_x F_{t,s})$$

With the Taylor approximation and the relation $F_{t,t} \approx v_t^*$ obtained from $\mathcal{L}_{\text{CFM}}$, we have

$$F_{t,s} = F_{t,t} + d \cdot \partial_s F_{t,t} + O(d^2) \approx v_t^* + d \cdot \partial_s F_{t,t} + O(d^2)$$

The identity $\partial_s F_{t,s} = \partial_s F_{t,t} + O(d)$ implies

$$v_t^* - F_{t,s}(x_t) = -d \cdot \partial_s F_{t,s} + O(d^2)$$

Hence,

$$f'(x_t) = d[\partial_t F_{t,s} + v_t^* \cdot \nabla_x F_{t,s} - \partial_s F_{t,s}] + O(d^2)$$

Since $v_t^* \approx F_{t,t} = F_{t,s} + O(d)$, we further obtain

$$f'(x_t) = d[\partial_t F_{t,s} + F_{t,s} \cdot \nabla_x F_{t,s} - \partial_s F_{t,s}] + O(d^2)$$

$$\implies \mathcal{L} = \mathbb{E}_{x,z,t,s} \left[ \|F_\theta(x_t; t, t) - v_t\|_2^2 + \frac{1}{4} \left\| \frac{d}{dt} f_\theta(x_t; t, s) \right\|_2^2 + O(d^3) \right]$$

We observe that there is a discrepancy between Eulerian distillation, expressed as $\|v_t - F_{t,s} + d(\partial_t F_{t,s} + v_t^* \cdot \nabla_x F_{t,s})\|_2^2$. The first $v_t^*$ term corresponds to $F_{t,t}$, while the second $v_t$ corresponds to $F_{t,s}$. Alternatively, since $F_{t,t} = F_{t,s} + O(d)$, both terms can be represented in terms of $F_{t,t}$.

In the case of $F_{t,t}$, the model learns $F_{t,t}^\theta(x_t) \approx v_t^*(x_t)$ due to the loss term of $\|F_{t,t}^\theta(x_t) - v_t(x_t|x)\|_2^2$. This can be interpreted as the model learning a flow map corresponding to the trajectory induced by an approximated marginal velocity.

Therefore, the Shortcut Model can be seen as Eulerian distillation under $O((s-t)^3)$-bound. Since Flow Map Models typically operate under the assumption $s \in [0, t)$, the Shortcut model's sampling scheme with $d \in [2^{-7}, 1]$ makes this error non-negligible. $\qquad\square$

**Consistency Flow Matching.** For linear interpolation $x_t = (1-t)x + tz$, define

$$f_\theta(x_t; t, s = 0) = x_t - tF_\theta(x_t; t, s = 0) \implies f_\theta(x_t; t) = x_t - tF_\theta(x_t; t)$$

Then, the Consistency Flow Matching objective from Yang et al. (2024) becomes

$$\mathcal{L}(\theta) = \mathbb{E}\left[ \|f_\theta(x_t; t) - f_{\theta^-}(x_{t-\Delta t}; t - \Delta t)\|_2^2 + \alpha\|F_\theta(x_t; t) - F_{\theta^-}(x_{t-\Delta t}; x - \Delta t)\|_2^2 \right]$$

The first term on the right side is the Taylor approximation of the consistency training objective, and the second term is the regularizer. Hence, we can interpret Consistency Flow Matching as a training flow map model via the approximation with regularization. $\qquad\square$

**UCGM.** For arbitrary interpolation of $\alpha_t, \sigma_t$, setting $s = 0$ yields

$$f_\theta(x_t; t) = \nu_t^{-1}(\sigma'_t x_t - \sigma_t F_\theta)$$

We can reformulate the objective while keeping the gradient unchanged:

$$\nabla_\theta \|f_\theta(x_t; t) - f_{\theta^-}(x_{\lambda t}; \lambda t)\|_2^2$$

$$= 2[\nabla_\theta f_\theta(x_t; t)]^T (t - \lambda t) \frac{f_{\theta^-}(x_t; t) - f_{\theta^-}(x_{\lambda t}; \lambda t)}{t - \lambda t}$$

$$\propto [\nabla_\theta f_\theta(x_t; t)]^T \frac{f_{\theta^-}(x_t; t) - f_{\theta^-}(x_{\lambda t}; \lambda t)}{t - \lambda t}$$

$$= \frac{\sigma_t}{\nu_t} [\nabla_\theta F_\theta(x_t; t)]^T \frac{f_{\theta^-}(x_t; t) - f_{\theta^-}(x_{\lambda t}; \lambda t)}{t - \lambda t}$$

$$= \nabla_\theta \left\| F_\theta(x_t; t) - F_{\theta^-}(x_t; t) + \frac{\sigma_t[f_{\theta^-}(x_t; t) - f_{\theta^-}(x_{\lambda t}; \lambda t)]}{\nu_t(t - \lambda t)} \right\|_2^2$$

which is identical to the objective of UCGM. When $\lambda = 0$, this reduces to the flow matching objective since $\lambda t = 0$ collapses $f_{\theta^-}(x_0; 0) = x_0$. In this case, the objective becomes origin

prediction, which in turn leads to $F_\theta$ with the velocity matching objective. Otherwise, setting $\lambda \to 1$ reduces the objective to consistency training by $\Delta = F_\theta(x_t; t) - F_{\theta^-}(x_t; t) + \frac{\sigma_t}{\nu_t}\frac{df_{\theta^-}(x_t;t)}{dt}$.

For $\lambda \in (0, 1)$, the objective $||f_\theta(x_t; t) - f_{\theta^-}(x_{\lambda t}; \lambda t)||_2^2$ yields consistency along the geometric sequence $\mathcal{T}_\lambda(t) = \{\lambda^k t\}_{k=0}^N$.

Define

$$g_t(x_t) = (x_t - \alpha_t f_t(x_t))\sigma_t^{-1}$$

Then, when $f_t(x_t) = x$, it follows that $g_t(x_t) = z$ for $x_t = \alpha_t x + \sigma_t z$. Using this, the flow map can be formulated in a DDIM-like manner as

$$f_{t,s}(x_t) = \alpha_s f_t(x_t) + \sigma_s g_t(x_t) = \frac{\sigma_s}{\sigma_t}x_t + (\alpha_s - \frac{\sigma_s}{\sigma_t}\alpha_t)f_t(x_t)$$

Assuming the composition chain

$$f_{s,r}(f_{t,s}(x_t)) = f_{s,r}(\tilde{x}_s) = \frac{\sigma_r}{\sigma_s}\tilde{x}_s + (\alpha_r - \frac{\sigma_r}{\sigma_s}\alpha_s)f_s(\tilde{x}_s)$$

for $\tilde{x}_s = f_{t,s}(x_t)$, we obtain

$$f_{t,r}(x_t) - f_{s,r}(f_{t,s}(x_t)) = (\alpha_r - \frac{\sigma_r}{\sigma_s}\alpha_s)(f_t(x_t) - f_s(\tilde{x}_s))$$

For $s = \lambda^k t$ for some $k \in \mathbb{N}$, if it follows that $\tilde{x}_s = f_{t,s}(x_t) \approx x_s$, then $f_s(x_s) = f_t(x_t)$ and $f_{t,r} = f_{s,r} \circ f_{t,s}$. In this case, the flow map can be constructed along the geometric sequence $\mathcal{T}_\lambda(t)$.

In general, the velocity of the DDIM map is given by $\frac{d}{ds}f_{t,s} = \alpha_s' f_t + \sigma_s' g_t$. Since the unconditional velocity is

$$v_t^*(x_t) = \alpha_t'\mathbb{E}_{x|x_t}[x] + \sigma_t'\mathbb{E}_{x|x_t}[(x_t - \alpha_t x)\sigma_t^{-1}],$$

the DDIM map coincides with the flow map only when $v_s^*(f_{t,s}(x_t)) = \frac{d}{ds}f_{t,s}(x_t)$, which implies $\mathbb{E}_{x|\tilde{x}_s}[x] = f_t(x_t)$. Setting $s \to t$ reduces this condition to $\mathbb{E}_{x|x_t}[x] = f_t(x_t)$ by the identity assumption. However, this condition fails to preserve the injectivity of the flow map at $t = 1$ due to the mean collapse problem, which leads to a contradiction. Therefore, the DDIM-style map does not coincide with the flow map in general. $\square$

**Reflow.** Rectified flows introduce Reflow to straighten trajectories after training. In Reflow, sampling from the trained model is performed via

$$x_0 = x_1 + \int_1^0 v_\theta(x_t; t)dt \approx \text{ODESolver}(v_\theta, x_1, 1, 0)$$

followed by the finetune w.r.t. the coupling $\Pi_{Z,\theta} = p_Z(z)p_{v_\theta}(x|z)$. The velocity $\hat{v}_t$ of the trajectory induced by the coupling $\Pi_{Z,\theta}$ is given by

$$\hat{v}_t = x_1 - \left(x_1 + \int_1^0 v_\theta(x_t; t)dt\right) = \int_0^1 v_\theta(x_t; t)dt$$

which corresponds to the displacement of the flow map. Therefore, Reflow can be interpreted as direct supervision of the flow map under linear interpolation.

A.6 SUBOPTIMALITY OF DIRECT TRAINING

Unlike Eulerian distillation, direct training does not guarantee convergence to the optimal flow map. Consider the direct training objective using the conditional velocity:

$$\mathcal{L}_{\text{DT}} = \mathbb{E}_{x,z,t,s}\left[\|\partial_t f_\theta(x_t; t, s) + v(x_t|x) \cdot \nabla_x f_\theta(x_t; t, s)\|_2^2\right]$$

By defining the velocity error as $\Delta v = v_t(x_t|x) - v_t^*(x_t)$, we can rewrite the objective in the form of Eulerian Distillation:

$$\mathcal{L}_{\text{DT}} = \mathbb{E}_{x,z,t,s}\left[\|\partial_t f_\theta(x_t; t, s) + (\Delta v + v_t^*(x_t)) \cdot \nabla_x f_\theta(x_t; t, s)\|_2^2\right]$$

$$= \mathbb{E}_{x,z,t,s}\left[\|\partial_t f_\theta(x_t; t, s) + \Delta v \cdot \nabla_x f_\theta(x_t; t, s) + v_t^*(x_t) \cdot \nabla_x f_\theta(x_t; t, s)\|_2^2\right]$$

$$= \mathbb{E}_{x,z,t,s}\left[\|\partial_t f_\theta(x_t; t, s) + v_t^*(x_t) \cdot \nabla_x f_\theta(x_t; t, s)\|_2^2\right] + \mathbb{E}_{x,z,t,s}\left[\|\Delta v \cdot \nabla_x f_\theta(x_t; t, s)\|_2^2\right]$$

by the law of total variance since other terms are independent of $x$, and $\mathbb{E}_{x|x_t}[\Delta v] = 0$. In this case, the second term can be represented as

$$\mathbb{E}_{x,z,t,s}\Big[\mathrm{Var}_{x|x_t}\left[\Delta v \cdot \nabla_x f_\theta(x_t; t, s)\right]\Big]$$

Under an independent coupling, the velocity error $\Delta v = v_t(x_t|x) - v_t^*(x_t)$ is typically nonzero. Consequently, unless $\|\nabla_x f_\theta(x_t; t, s)\|$ collapses to zero, the objective function inherently contains a larger variance term compared to that of Eulerian distillation.

To minimize the overall loss, the optimizer faces a trade-off involving this variance. This introduces a bias that distorts the learned flow map towards becoming flatter by an external force $\Delta v_t = v_t - v_t^* \perp \nabla_x f_\theta(x_t; t, s)$. Therefore, direct training is not guaranteed to converge to the ground-truth flow map due to this distorting variance term.

In flow matching, even when the loss term $\mathbb{E}_{x,z,t}[\|v(x_t|x) - F_\theta(x_t; t)\|_2^2]$ is decomposed as follows

$$\mathbb{E}\Big[\|\Delta v + v^*(x_t) - F_\theta(x_t; t)\|_2^2\Big] = \mathbb{E}\Big[\|v^*(x_t) - F_\theta(x_t; t)\|_2^2\Big] + \mathrm{Var}[\Delta v]$$

The variance term, $\mathrm{Var}[\Delta v]$, is independent of the network. Therefore, it does not affect convergence to the global optimum. $\qquad\square$

**Euler-Lagrange Equation.** To find the optimum of the direct training objective, we apply the Euler-Lagrange equation. The objective can be represented in the vector form as

$$\mathcal{L}_{\mathrm{DT}} = \mathbb{E}_{x,z,t,s}\left[\|\partial_t f_{t,s}(x_t) + v(x_t|x)^T \nabla_x f_{t,s}(x_t)\|_2^2\right] = \iint_\Omega \rho_t(x_t)\mathbb{E}_{x,z,s|x_t}[\|\cdot\|_2^2]dx_t dt$$

We set the conditional expectation as the Lagrangian,

$$L(f, \partial_t f, \nabla_x f) = \mathbb{E}_{x,z,s|x_t}[\|\partial_t f_{t,s}(x_t) + v(x_t|x)^T \nabla_x f_{t,s}(x_t)\|_2^2]$$

The corresponding Euler-Lagrange equation is

$$\frac{\partial L}{\partial f} - \partial_t\left(\frac{\partial L}{\partial(\partial_t f)}\right) - \nabla_x\left(\frac{\partial L}{\partial(\nabla_x f)}\right) = 0 \iff \mathbb{E}_{x,z,s|x_t}[\partial_t R + \nabla_x \cdot (vR)] = 0$$

where the residue is defined as

$$R = \partial_t f_{t,s}(x_t) + v(x_t|x)^T \nabla_x f_{t,s}(x_t)$$

Letting $\Delta v = v(x_t|x) - v_t^*(x_t)$ and introducing the operator $L_* f = \partial_t f + (v_t^*)^T \nabla_x f$, the residue can be rewritten as $R = L_* f_{t,s} + \Delta v^T \nabla_x f_{t,s}$, so that

$$\mathbb{E}_{x|x_t}[\partial_t R] = \partial_t \mathbb{E}_{x|x_t}[R] = \partial_t(L_* f_{t,s})$$
$$\mathbb{E}_{x|x_t}[\nabla_x \cdot (vR)] = \nabla_x \cdot \mathbb{E}_{x|x_t}[(v^* + \Delta v)(L_* f_{t,s} + \Delta v^T \nabla_x f_{t,s})]$$
$$= \nabla_x \cdot \left(v^* L_* f_{t,s} + \mathbb{E}_{x|x_t}[\Delta v(\Delta v^T \nabla_x f_{t,s})]\right)$$
$$= \nabla_x \cdot \left(v^* L_* f_{t,s} + \Sigma_{\Delta v|x_t} \nabla_x f_{t,s}\right)$$

Therefore, the optimality condition becomes

$$\mathcal{EL} = \mathbb{E}_{x,z,s|x_t}[\partial_t R + \nabla_x \cdot (vR)]$$
$$= \partial_t(L_* f_{t,s}) + \nabla_x \cdot (v_t^* L_* f_{t,s}) + \nabla_x \cdot (\Sigma_{\Delta v|x_t} \nabla_x f_{t,s}) = 0$$

If we assume the $L^2$-adjoint of $L_*$ to be $L^* f = -\partial_t f - \nabla \cdot \left((v_t^*)^T f\right)$, the condition simplifies to

$$L^* L_* f_{t,s} - \nabla_x \cdot (\Sigma_{\Delta v|x_t} \nabla_x f_{t,s}) = 0 \iff \|L_* f\|_2^2 + \nabla_x \cdot (\Sigma_{\Delta v|x_t} \nabla_x f_{t,s}) = 0$$

Thus, the optimum of the direct training arises precisely when the above condition is satisfied. When $\Sigma_{\Delta v|x_t} = \mathrm{Cov}_{x|x_t}[v(x_t|x)] \to 0$, the condition reduces to $L_* f_{t,s} = 0$, which is equivalent to Eulerian distillation. In this case, the quadratic structure guarantees convergence through PSD curvature at the global optimum. Otherwise, due to the covariance term, the condition cannot be written in the form $\partial_t f + w_t \cdot \nabla_x f$, and hence no single drift can consistently drive the flow map. $\quad\square$

### A.7 INSTABILITY OF CONSISTENCY TRAINING

The continuous-time consistency training objective employs a stop-gradient operation, ensuring that the main objective remains unchanged while making computation more efficient. In this case, the objective is defined as

$$\mathcal{L}_{\text{CT}} = \mathbb{E}_{x,z,t,s} \left[ f_{t,s}(x_t)^T \frac{df_{t,s}^-(x_t)}{dt} \right]$$

where $f_{t,s}(x_t) = f_\theta(x_t; t, s)$ and detaching gradient is denoted by $f_{t,s}^-(x_t) = f_{\theta^-}(x_t; t, s)$.

$\mathcal{L}_{\text{CT}}$ reduces in value to Eulerian distillation even along the conditional trajectory, as demonstrated in Appendix A.5, paragraph on **sCT**. However, since the objective is expressed as a linear term, the Euler-Lagrange equation cannot determine a stationary point, as it contains no explicit terms of $f_{t,s}$ unless the gradient is detached:

$$\mathcal{EL} = \mathbb{E}_{x,z,s|x_t} \left[ L f_{t,s}^- + v_t^T \nabla_x f_{t,s}^- \right] = 0$$

Moreover, while the quadratic term in Eulerian distillation ensures PSD curvature and provides stable convergence at the optimum, the consistency training objective does not guarantee convergence, as the Hessian vanishes and the curvature required to stabilize the optimum is absent. It only specifies the fixed point on $L f_{t,s} = 0$, and the gradient dynamics alone may fail to converge. $\qquad\square$

### A.8 SUBOPTIMALITY OF NETWORK-INDUCED COUPLING

For an arbitrary coupling $(\hat{x}, \hat{z}) \sim \Pi_{X,Z}$ with $\hat{x}_t = \alpha_t \hat{x} + \sigma_t \hat{z}$, the gap between the conditional and marginal velocities is given by

$$\Delta v = \hat{x}_t' - v_t^*(\hat{x}_t) = \alpha_t'(\hat{x} - \mu_{x|\hat{x}_t}) + \sigma_t'(\hat{z} - \mu_{z|\hat{x}_t})$$

where $\mu_{x|x_t} = \mathbb{E}_{x|x_t}[x]$ and $\mu_{z|x_t} = \mathbb{E}_{z|x_t}[z]$ are conditional means. This follows since $v_t^*(x_t) = \mathbb{E}_{x|x_t}[v(x_t|x)]$ can be expressed as $\mathbb{E}_{x,z|x_t}[\alpha_t' x + \sigma_t' z]$. The general form of the loss can be expressed by

$$\mathbb{E} \left[ \| A + g^T \Delta v \|_2^2 \right] = \mathbb{E} \left[ \| A \|_2^2 \right] + 2\mathbb{E}[A^T g^T \Delta v] + \mathbb{E}[\| g^T \Delta v \|_2^2]$$

where $A = \partial_t f_{t,s}^\theta + g^T v_t^*(x_t)$ and $g = \nabla_x f_{t,s}^\theta$.

**Case 1: Independent Coupling.** In this setup, we use an independent coupling $(x, z) \sim p_{\text{data}} \times p_Z$ and the conditional velocity $v_t(x_t|x)$, which corresponds to setting $\hat{x} = x$ and $\hat{z} = z$. The velocity gap is $\Delta v = v_t(x_t|x) - v_t^*(x_t)$. The cross-term vanishes because the expectation of the gap is zero conditioned on $x_t$:

$$\mathbb{E}_{x,z|x_t}[\Delta v] = \mathbb{E}_{x,z|x_t}[v_t(x_t|x) - v_t^*(x_t)] = v_t^*(x_t) - v_t^*(x_t) = 0$$

Thus, $2\mathbb{E}[A \Delta v \cdot g] = 0$ and the loss simplifies to

$$\mathcal{L}_{\text{IC}} = \mathcal{L}_{\text{ED}} + \mathbb{E}[\| g^T \Delta v \|_2^2] = \mathcal{L}_{\text{ED}} + \mathbb{E} \left[ \text{Var}_{x|x_t}[g^T \Delta v] \right]$$

The variance term, which represents the error from Eulerian distillation, can be expanded as:

$$\mathcal{E}_{\text{IC}} = \text{Var}_{x,z|x_t} \left[ g^T \left( \alpha_t'(x - \mu_{x|x_t}) + \sigma_t'(z - \mu_{z|x_t}) \right) \right]$$
$$= (\alpha_t')^2 g^T \Sigma_{x|x_t} g + (\sigma_t')^2 g^T \Sigma_{z|x_t} g + 2\alpha_t' \sigma_t' g^T \Sigma_{xz|x_t} g$$

where $\Sigma_{x|x_t} = \text{Cov}_{x|x_t}(x, x)$, $\Sigma_{z|x_t} = \text{Cov}_{z|x_t}(z, z)$, and $\Sigma_{xz|x_t} = \text{Cov}_{x,z|x_t}(x, z)$.

**Case 2: Generator-Induced Coupling.** Generator-induced coupling methods replace one of the variables with a network prediction, $\hat{x} = f_{t,0}^\theta(x_t)$ with stop-gradient: $\hat{x}_t = \alpha_t f_{t,0}(x_t) + \sigma_t z$. For the first case, the velocity gap is:

$$\Delta v = \alpha_t'(f_{t,0}(x_t) - \mu_{x|\hat{x}_t}) + \sigma_t'(z - \mu_{z|\hat{x}_t})$$

The conditional expectation of the gap is no longer zero in general:

$$\mathbb{E}_{z|x_t}[\Delta v] = \alpha_t' \left( f_{t,0}^\theta(x_t) - \mu_{x|\hat{x}_t} \right)$$

This introduces a non-zero cross-term in the loss, then the total error term for GC is:

$$\mathcal{E}_{\text{GC}} = 2\alpha_t' A^T g^T (f_{t,0}^\theta(x_t) - \mu_{x|\hat{x}_t}) + (\alpha_t')^2 [g^T (f_{t,0}(x_t) - \mu_{x|\hat{x}_t})]^2 + (\sigma_t')^2 g^T \Sigma_{z|x_t} g$$

Comparing the error terms, we find that $\mathcal{E}_{\text{IC}} > \mathcal{E}_{\text{GC}}$ if:

$$(\alpha_t')^2 g^T \Sigma_{x|x_t} g + 2\alpha_t' \sigma_t' g^T \Sigma_{xz|x_t} g > 2\alpha_t' A^T g^T (f_{t,0}^\theta(x_t) - \mu_{x|\hat{x}_t}) + (\alpha_t')^2 [g^T (f_{t,0}^\theta(x_t) - \mu_{x|\hat{x}_t})]^2$$

When the generator is a good estimator of the posterior mean, i.e., $f_{t,0}^\theta(x_t) \approx \mu_{x|\hat{x}_t}$, the right side becomes small. However, at $t = 1$, this condition reduced to $f_{1,0}^\theta(z) = \mu_{x|z} = \mu_X$, as shown in A.1, which results in posterior mean collapse. In this case, $f_{1,0}^\theta$ becomes constant as $\mathcal{E}_{\text{GC}}$ approaches zero, violating the injectivity required for a well-defined flow map, leading to a contradiction. Thus, generator-induced coupling reduces but cannot eliminate the error, preventing guaranteed convergence. It is also a suboptimal choice when continuous-time consistency training is employed.

Silvestri et al. (2025) introduces an additional network $g : x \mapsto z$ for $\hat{x} = x$ and $\hat{z} = g(x)$. In this case, enforcing $g(x) \approx \mu_{z|x}$ can reduce the gap, but does not eliminate the whole, since the term $(\alpha_t')^2 g^T \Sigma_{x|x_t} g$ remains. From a consistency training perspective, since the independent coupling already exhibits a marginal velocity field, this becomes a negligible choice with respect to guaranteeing marginal velocity; however, it can reduce the variance of the loss when $g$ is a good posterior approximator. $\qquad\square$

### A.9 LINEARIZATION COST

Recall that $f_{t,s}^\theta(x_t) = \nu^{-1}(A_{t,s}' x_t - A_{t,s} F_{t,s}^\theta(x_t))$ for $A_{t,s} = \alpha_s \sigma_t - \sigma_s \alpha_t$. Differentiating with respect to timestep $t$ gives

$$\frac{df_\theta(x_t; t, s)}{dt} = \nu^{-1}\left( A_{t,s}'' x_t + A_{t,s}'(v_t^*(x_t) - F_\theta(x_t; t, s)) - A_{t,s}\frac{dF_\theta(x_t; t, s)}{dt} \right)$$

Following Lu & Song (2025), the gradient of the Eulerian distillation can be written as

$$\nabla_\theta \mathbb{E}\left[ 2 f_\theta^T(x_t; t, s)\frac{df_{\theta^-}(x_t; t, s)}{dt} \right]$$

$$\propto \nabla_\theta \mathbb{E}\left[ -A_{t,s}\nu^{-2} F_\theta(x_t; t, s) \cdot \left( A_{t,s}'' x_t + A_{t,s}'(v_t^*(x_t) - F_{\theta^-}(x_t; t, s)) - A_{t,s}\frac{dF_{\theta^-}(x_t; t, s)}{dt} \right) \right]$$

$$= A_{t,s}\nu^{-2} \cdot \nabla_\theta \mathbb{E}\left[ \|F_\theta(x_t; t, s) - \text{sg}[F_{\text{tgt}}(x_t; t, s)]\|_2^2 \right]$$

where $F_{\text{tgt}}(x_t; t, s) = F_\theta(x_t; t, s) + \left( A_{t,s}'' x_t + A_{t,s}'(v_t^*(x_t; t) - F_\theta(x_t; t, s)) - A_{t,s}\frac{dF_\theta(x_t; t, s)}{dt} \right)$

In this case, $v_t^* - F_\theta$ can be interpreted as the flow matching term, and $dF_\theta/dt$ as a linearization term involving the JVP, which penalizes the $t$-dependent outputs of $f_\theta$. For a linear interpolation, $A_{t,s}$ takes the form $A_{t,s} = t - s$, while for a trigonometric interpolation $A_{t,s} = \sin(t - s)$, both are proportional to $t - s$. Their derivatives are $A_{t,s}' = 1$ and $A_{t,s}' = \cos(t - s)$, respectively.

As $s \to t$ and $(t - s) \to 0$, the contribution of the linearization term vanishes , while the flow matching term is amplified. Conversely, as $s \to 0$, the linearization term is amplified and the flow matching term diminishes.

We note that the linearization cost increases with step size, making optimization more challenging. This is because the linearization term involves more the complex structure given by the JVP, while the flow matching term requires only simple forward pass.

## B IMPROVED SELF-DISTILLATION

### B.1 GUARANTEE THE CONVERGENCE

Revisit our objective

$$\mathcal{L} = \mathbb{E}_{x,z,t,s}\left[ \left\| F_{t,t}^\theta(x_t; t, t) - v_t(x_t|x) \right\|_2^2 + \left\| \partial_t f_{t,s}^\theta(x_t) + F_{t,t}^{\theta^-}(x_t) \cdot \nabla_x f_{t,s}^\theta(x_t) \right\|_2^2 \right]$$

The first term of the right side trains $F_{t,t}^\theta(x_t)$ to approximate the marginal velocity via the flow matching objective, while the second term learns the flow map $f_{t,s}$ along the trajectory of $F_{t,t}^{\theta^-}$ in a self-distillation manner.

Individually, each term is guaranteed to converge to its desired optimum, the marginal velocity and the flow map of the velocity $F_{t,t}^{\theta^-}$ by Prop. 3.1 (Appendix A.3). From a joint perspective, we need to consider $t = s$, since the network is forced to optimize both terms simultaneously at this point. As Eulerian distillation collapses to the flow matching objective when $t \to s$, the second term trains the model to learn the instantaneous velocity of the trajectory (Appendix A.9). In this case, $F_{t,t}^\theta$ learns from $F_{t,t}^{\theta^-}$, and inductively approximates $v_t^*(x_t)$ through the first term. This naturally reduces to a non-conflict joint training. For $t \neq s$, the network is conditioned differently in the two terms, and it can learn the proper mapping provided that the network capacity is sufficient.

Consequently, the overall objective trains the network to follow the marginal velocity as the trajectory of the flow map naturally. $\qquad\square$

## B.2 Deriving Final Objective

Recall the consistency training objective under the generalized flow map (Appendix. A.9):

$$A_{t,s}\nu^{-2} \cdot \mathbb{E}\left[\|F_\theta(x_t; t, s) - \mathrm{sg}[F_{\mathrm{tgt}}(x_t; t, s)]\|_2^2\right]$$

where $F_{\mathrm{tgt}}(x_t; t, s) = F_\theta(x_t; t, s) + \left(A_{t,s}'' x_t + A_{t,s}'(v_t^*(x_t; t) - F_\theta(x_t; t, s)) - A_{t,s}\frac{dF_\theta(x_t; t, s)}{dt}\right)$.

To follow the marginal velocity, we replace $v_t^*(x_t)$ with instantaneous velocity $F_{t,t}^\theta(x_t)$ while jointly training with $\mathcal{L}_{\mathrm{CFM}}$.

Particularly, for linear interpolation, we have $A_{t,s} = t - s$, $A_{t,s}' = 1$ and $A_{t,s}'' = 0$. This simplifies the target to $F_{\mathrm{tgt}}^{\mathrm{lin}}(x_t; t, s) = v_t^*(x_t) - (t-s)F_\theta'(x_t; t, s)$ which coincides with the regression target of MeanFlow. For trigonometric interpolation, we have $A_{t,s} = \sin(t-s)$, $A_{t,s}' = \cos(t-s)$ and $A_{t,s}'' = -\sin(t-s)$. Thus, the target becomes $F_{\mathrm{tgt}}^{\mathrm{tri}} = F_\theta(x_t; t, s) + \cos(t-s) \cdot (v_t^*(x_t) - F_\theta(x_t; t, s)) - \sin(t-s) \cdot (x_t + F_\theta'(x_t; t, s))$.

Although consistency training already guarantees the marginal flow map at its fixed point, the gradient in practice can exhibit a gap expressed as:

$$\mathbb{E}_{x,z,t,s}[f_{t,s}^\theta(\Delta v \cdot \nabla_x f_{t,s}^{\theta^-})]$$

When self-distillation is combined with flow matching, $\mathbb{E}\left[\|F_\theta(x_t; t, t) - v(x_t|x)\|_2^2\right]$, the velocity error $\Delta v = F_\theta(x_t; t, t) - v_t^*(x_t)$ can be further reduced compared to $\Delta v = v_t(x_t|x) - v_t^*(x_t)$, thereby stabilizing the training.

To incorporate classifier-free guidance in the subsequent discussion, we set $v_\theta(x_t; t)$ to the approximated marginal velocity, as an alternative to $F_{t,t}^\theta(x_t)$.

For JVP approximation, to ensure that $dx_t/dt$ follows the velocity $v_\theta(x_t; t)$, we approximate

$$\frac{dx_t}{dt} \approx \frac{[x_t + \epsilon \cdot v_\theta(x_t; t)] - [x_t - \epsilon \cdot v_\theta(x_t; t)]}{2\epsilon} = v_\theta(x_t; t)$$

Thus, the full JVP approximation becomes

$$F_\theta'(x_t; t, s) = \frac{dF_\theta(x_t; t, s)}{dt} = \frac{F_\theta(x_t + \epsilon \cdot v_\theta(x_t; t), t + \epsilon, s) - F_\theta(x_t - \epsilon \cdot v_\theta(x_t; t), t - \epsilon, s)}{2\epsilon}$$

Applying adaptive weighting, our final objective is

$$\tilde{\mathcal{L}}_{t,s}(x_t, x) = \|F_\theta(x_t; t, t) - v_t(x_t|x)\|_2^2 + \|F_\theta(x_t; t, s) - \mathrm{sg}[F_{\mathrm{tgt}}(x_t; t, s)]\|_2^2$$

$$\mathcal{L}_{\mathrm{iSD}} = \mathbb{E}_{x,z,t,s}\left[w_{t,s}(x_t, x) \cdot \tilde{\mathcal{L}}_{t,s}(x_t, x)\right]; \text{ where } w_{t,s}(x_t, x) = (\mathrm{sg}[\tilde{\mathcal{L}}_{t,s}(x_t, x)] + \eta)^{-p}$$

Detailed training and sampling algorithms are provided in Alg. 2 and Alg. 3.

---

**Algorithm 1 (Time Sampler)** Timestep sampling function using beta distribution

---

1: **function** time_sampler$(\theta_1, \theta_2)$
2: $\quad t, s \sim \text{Beta}(\theta_1, \theta_2)$ $\qquad\qquad\qquad\qquad\qquad\qquad\qquad\qquad\qquad$ ▷ Timestep Sampling
3: $\quad$ **return** $\frac{\pi}{2}\max(t,s), \frac{\pi}{2}\min(t,s)$

---

**Algorithm 2 (iSD Training)** Training algorithm of vanilla iSD

---

**Require:** Noise distribution $p_Z$, data distribution $p_X$, model $F_\theta$, learning rate $\mu$, time distribution
$\quad (\theta_1, \theta_2)$, adaptive weighting $(\eta, p)$, JVP approximation step size $\epsilon$, class labels $c$
**Ensure:**
1: **repeat**
2: $\quad z \sim p_Z,\ x \sim p_X$
3: $\quad t, s \leftarrow$ time_sampler$(\theta_1, \theta_2)$
4: $\quad x_t \leftarrow \cos(t)x + \sin(t)z, \quad v_t \leftarrow \cos(t)z - \sin(t)x$
5: $\quad F_{t,s} \leftarrow F_\theta(x_t; t, s, c), \qquad F_{t,t} \leftarrow F_\theta(x_t; t, t, c)$
6: $\quad F'_{t,s} \leftarrow \left[F_\theta(x_t + \epsilon F_{t,t}; t+\epsilon, s, c) - F_\theta(x_t - \epsilon F_{t,t}; t-\epsilon, s, c)\right]/(2\epsilon)$ $\qquad$ ▷ JVP-Approx.
7: $\quad F_{\text{tgt}} \leftarrow F_{t,s} + \cos(t-s) \cdot (F_{t,t} - F_{t,s}) - \sin(t-s) \cdot (x_t + F'_{t,s})$
8: $\quad \tilde{\mathcal{L}}_{t,s} \leftarrow \|F_{t,t} - v_t\|_2^2 + \|F_{t,s} - \text{sg}[F_{\text{tgt}}]\|_2^2$ $\qquad\qquad\qquad$ ▷ Optimization Target
9: $\quad \mathcal{L} \leftarrow \tilde{\mathcal{L}}_{t,s} \times (\text{sg}[\tilde{\mathcal{L}}_{t,s}] + \eta)^{-p}$ $\qquad\qquad\qquad\qquad\qquad$ ▷ Adaptive Weighting
10: $\quad \theta \leftarrow \theta - \mu\nabla_\theta\mathcal{L}$ $\qquad\qquad\qquad\qquad\qquad\qquad\qquad$ ▷ Model Update
11: **until** Convergence

---

**Algorithm 3 (iSD Sampling)** Sampling algorithm of vanilla iSD

---

**Require:** Initial noise $z \sim p_Z$, trained model $F_\theta$, class labels $c$, sampling time steps $\{t_i\}_{i=1}^N$
**Ensure:**
1: $x \leftarrow z$
2: **for** $i \leftarrow 1$ to $N$ **do**
3: $\quad x \leftarrow \cos(t_{i+1} - t_i) \cdot x + \sin(t_{i+1} - t_i) \cdot F_\theta(x; t_i, t_{i+1}, c)$
4: **end for**
5: **return** $x$

---

### B.3 CLASSIFIER-FREE GUIDANCE OF FLOW MAP MODELS

By abstracting the guiding trajectory to $v_\theta$, the flow map model can naturally be trained to follow the specific trajectory as long as it is Lipschitz continuous. Given a label $c$ and an null class label $\varnothing$, let the corresponding velocity fields be $F_{t,t}(x_t; c)$ and $F_{t,t}(x_t; \varnothing)$. If both are globally Lipschitz continuous, then the CFG trajectory $\tilde{v}_\theta(x_t; t, c) = \tilde{v}_t(x_t; c)$ is also globally Lipschitz continuous, since any linear combination of Lipschitz continuous functions remains Lipschitz continuous.

$$\tilde{v}_\theta(x_t; t, c) = F_\theta(x_t; t, t, \varnothing) + \omega(F_\theta(x_t; t, t, c) - F_\theta(x_t; t, t, \varnothing))$$

Thus, the flow map can be trained to follow the CFG velocity field, referred to as *Pre-CFG*. In this case, we need to address a conflict: $\mathcal{L}_{\text{CFM}}$ enforces $F_{t,t} \approx v_t^*$ while $\mathcal{L}_{\text{SD-R}}$ enforces $F_{t,t} \approx \tilde{v}_t$. To resolve this, we append the guidance scale $\omega$ as an additional condition, $F_{t,t}^\theta(x_t; c, \omega)$. Then, the modified objectives are given by:

$$\mathcal{L}_{\text{CFM}} = \mathbb{E}\left[\|F_\theta(x_t; t, t, c, 1.0) - v_t(x_t|x)\|_2^2\right]$$

$$\mathcal{L}_{\text{SD-C}} = \mathbb{E}\left[\|F_\theta(x_t; t, s, c, \omega) - \text{sg}[F_{\text{tgt}}(x_t; t, s, c, \omega)]\|_2^2\right]$$

where $F_{\text{tgt}} = F_{t,s}(x_t; c, \omega) + \left(A''_{t,s}x_t + A'_{t,s}(\tilde{v}_t(x_t; c) - F_{t,s}(x_t; c, \omega)) - A_{t,s}F'_{t,s}(x_t; c, \omega)\right)$ and $\tilde{v}_t(x_t; c) = F_{t,t}(x_t; \varnothing, 1.0) + \omega(F_{t,t}(x_t; c, 1.0) - F_{t,t}(x_t; \varnothing, 1.0))$ with $\theta$ omitted for brevity. Detailed procedures follow Alg. 4 for $\mathcal{L}_{\text{iSD-U}}$ and Alg. 5 for $\mathcal{L}_{\text{iSD-C}}$.

Hence, $\mathcal{L}_{\text{CFM}}$ ensures $F_{t,t}(x_t; c, 1.0) \approx v_t^*(x_t; c)$, while $\mathcal{L}_{\text{SD-R}}$ ensures $F_{t,t}(x_t; c, \omega) \approx \tilde{v}_t(x_t; c)$. This choice is natural, as $\tilde{v}_t = v_t^*$ when $\omega = 1$.

However, *Post-CFG* defined as

$$\tilde{F}_\theta(x_t; t, s, c) = F_\theta(x_t; t, s, \varnothing) + \omega(F_\theta(x_t; t, s, c) - F_\theta(x_t; t, s, \varnothing))$$

does not follow the CFG trajectory. This discrepancy arises from the definition of the flow map,

$$f_\theta(x_t; t, s) = x_t + \int_t^s v_\tau^*(x_\tau) d\tau$$

which performs the path integral along a specific trajectory induced by $v_\tau^*$. For a CFG trajectory, the path integral should be taken along $\tilde{v}_\tau$. In contrast, Post-CFG computes two separate forward passes, integrating along $v_\tau^*(x_\tau; c)$ and $v_\tau^*(x_\tau; \varnothing)$, rather than along $\tilde{v}_\tau$. As a result, the integration differ from the expected CFG trajectory. The detailed procedure of Post-CFG follows Alg. 6.

---

**Algorithm 4 (iSD-U Training)** Training algorithm of iSD-U

---

**Require:** Noise distribution $p_Z$, data distribution $p_X$, model $F_\theta$, learning rate $\mu$, time distribution $(\theta_1, \theta_2)$, adaptive weighting $(\eta, p)$, JVP approximation $\epsilon$, Pre-CFG scale $\omega$, class labels $c$.
**Ensure:**
1: **repeat**
2:    $z \sim p_Z$, $x \sim p_X$
3:    $t, s \leftarrow \texttt{time\_sampler}(\theta_1, \theta_2)$
4:    $x_t \leftarrow \cos(t)x + \sin(t)z$,    $v_t \leftarrow \cos(t)z - \sin(t)x$
5:    $F_{t,s} \leftarrow F_\theta(x_t; t, s, c)$,       $F_{t,t} \leftarrow F_\theta(x_t; t, t, c)$
6:    $\tilde{v}_t = (1 - \omega)F_\theta(x_t; t, t, \varnothing) + \omega F_{t,t}$
7:    $F'_{t,s} \leftarrow [F_\theta(x_t + \epsilon \tilde{v}_t; t + \epsilon, s, c) - F_\theta(x_t - \epsilon \tilde{v}_t; t - \epsilon, s, c)] / (2\epsilon)$    ▷ JVP-Approx.
8:    $F_{\text{tgt}} \leftarrow F_{t,s} + \cos(t - s) \cdot (\tilde{v}_t - F_{t,s}) - \sin(t - s) \cdot (x_t + F'_{t,s})$
9:    $\tilde{\mathcal{L}}_{t,s} \leftarrow \|F_{t,t} - v_t\|_2^2 + \|F_{t,s} - \text{sg}[F_{\text{tgt}}]\|_2^2$    ▷ Optimization Target
10:   $\mathcal{L} \leftarrow \tilde{\mathcal{L}}_{t,s} \times (\text{sg}[\tilde{\mathcal{L}}_{t,s}] + \eta)^{-p}$    ▷ Adaptive Weighting
11:   $\theta \leftarrow \theta - \mu \nabla_\theta \mathcal{L}$    ▷ Model Update
12: **until** Convergence

---

**Algorithm 5 (iSD-C Training)** Training algorithm of iSD-C

---

**Require:** Noise distribution $p_Z$, data distribution $p_X$, model $F_\theta$, learning rate $\mu$, time distribution $(\theta_1, \theta_2)$, adaptive weighting $(\eta, p)$, JVP approximation $\epsilon$, Pre-CFG scale $\omega$, class labels $c$
**Ensure:**
1: **repeat**
2:    $z \sim p_Z$, $x \sim p_X$
3:    $t, s \leftarrow \texttt{time\_sampler}(\theta_1, \theta_2)$
4:    $x_t \leftarrow \cos(t)x + \sin(t)z$,       $v_t \leftarrow \cos(t)z - \sin(t)x$
5:    $F_{t,s,\omega} \leftarrow F_\theta(x_t; t, s, c, \omega)$,    $F_{t,t,\omega} \leftarrow F_\theta(x_t; t, t, c, \omega)$
6:    $F_{t,t,1.0} \leftarrow F_\theta(x_t; t, t, c, 1.0)$
7:    $\tilde{v}_t = (1 - \omega)F_\theta(x_t; t, t, \varnothing, 1.0) + \omega F_{t,t,1.0}$
8:    $F'_{t,s,\omega} \leftarrow [F_\theta(x_t + \epsilon \tilde{v}_t; t + \epsilon, s, c, \omega) - F_\theta(x_t - \epsilon \tilde{v}_t; t - \epsilon, s, c, \omega)] / (2\epsilon)$
9:    $F_{\text{tgt}} \leftarrow F_{t,s,\omega} + \cos(t - s) \cdot (\tilde{v}_t - F_{t,s,\omega}) - \sin(t - s) \cdot (x_t + F'_{t,s,\omega})$
10:   $\tilde{\mathcal{L}}_{t,s} \leftarrow \|F_{t,t,1.0} - v_t\|_2^2 + \|F_{t,s,\omega} - \text{sg}[F_{\text{tgt}}]\|_2^2$    ▷ Optimization Target
11:   $\mathcal{L} \leftarrow \tilde{\mathcal{L}}_{t,s} \times (\text{sg}[\tilde{\mathcal{L}}_{t,s}] + \eta)^{-p}$    ▷ Adaptive Weighting
12:   $\theta \leftarrow \theta - \mu \nabla_\theta \mathcal{L}$    ▷ Model Update
13: **until** Convergence

---

**Algorithm 6 (Post-CFG Sampling)** Sampling algorithm of iSD with Post-CFG

---

**Require:** Initial noise $z \sim p_Z$, model $F_\theta$, Post-CFG scale $\omega$, class labels $c$, sampling steps $\{t_i\}_{i=1}^N$
**Ensure:**
1: $x \leftarrow z$
2: **for** $i \leftarrow 1$ to $N$ **do**
3:    $\tilde{F}_{t,s} \leftarrow (1 - \omega)F_\theta(x; t_i, t_{i+1}, \varnothing) + \omega F_\theta(x; t_i, t_{i+1}, c)$    ▷ Post-CFG
4:    $x \leftarrow \cos(t_{i+1} - t_i) \cdot x + \sin(t_{i+1} - t_i) \cdot \tilde{F}_{t,s}$
5: **end for**
6: **return** $x$

## C    Experimental Details

### C.1    Reproducibility of Consistency Training

To evaluate the reproducibility of consistency training, we conducted experiments within the UCGM (Sun et al., 2025) framework. We compared the FID scores of several models trained under different initialization conditions. Following UCGM, we first extract latent representations of ImageNet-1K $256 \times 256$ using VAVAE (Yao et al., 2025). All models were trained with the same hyperparameters and settings: RAdam optimizer with a learning rate of 0.0001, weight decay of 0.0, $\beta_1 = 0.9$, $\beta_2 = 0.999$, batch size of 1024, gradient clipping at 0.1, and timestep $t$ sampled from Beta(0.8, 1.0). For enhancement, we applied a label drop ratio of 0.1, an enhancement range of $(0, 0.75)$, and an enhancement ratio of 2.0. We also used the cosine function as the loss weighting function and trained all models with linear interpolation for 40K iterations. Different experimental details are provided below.

**Multistep Baseline**    We trained the DiT-XL/1 architecture initialized from the publicly released multistep checkpoint of UCGM. This configuration achieved a 2-step FID of 2.69, which is reasonable but still falls short of the reported FID of 1.42.

**LightningDiT**    We trained the LightningDiT-XL/1 architecture from its released pretrained model. In this setting, the model achieved a 2-step FID of 10.01, which is worse than the reported FID.

**Reproduced Multistep Model**    In this experiment, we trained the DiT-XL/2 architecture from scratch to reproduce the multistep baseline. For training, we used AdamW (Loshchilov & Hutter, 2019) with a learning rate of 0.0002, $\beta_1 = 0.9$, $\beta_2 = 0.95$, EMA decay weight of 0.999, and timestep $t$ sampled from Beta(1, 1). We used an enhancement ratio of 0.47 and a cosine weighting function. After training the multistep baseline model for 800k iterations, we trained a few-step model initialized from the reproduced multistep baseline using consistency training with the same few-step settings. This resulted in a 2-step FID of 5.96.

**Without Preconditioner**    We train a DiT-XL/1 architecture from randomly initialized weights without any preconditioner. In this case, training consistently failed, with the loss diverging and no meaningful samples being generated. While 40K steps may appear insufficient for scratch training, other scratch training methods already show a rapid decrease by 40K steps. We consider that this is enough to check the unstable dynamics compared to other models.

These experiments suggest that consistency training is highly sensitive to initialization and the choice of preconditioner. It proves unstable under random initialization and requires a well-trained multistep baseline for stable optimization. Furthermore, even when initialized from a pretrained model, consistency training demonstrates limited robustness and reproducibility across different architectures and setups.

### C.2    Implementation

**ImageNet 256$\times$256**    SD-VAE (Rombach et al., 2022) was used to encode images from the ImageNet 256 dataset into a $32 \times 32 \times 4$ latent representation. For DiT (Peebles & Xie, 2023) models, we employed RMSNorm (Zhang & Sennrich, 2019), SiLU activation, QK normalization (Henry et al., 2020), and RoPE (Su et al., 2023) for minor improvements. Each model was scaled by depth and hidden dimension, while the patch size was kept fixed. We sampled $t$ and $s$ independently from Beta(0.8, 1.0), and set $t, s := \max(t, s), \min(t, s)$. For generations, we have simply sampled the time intervals uniformly without additional engineering. Detailed training parameters are provided in Tab. 6.

**CIFAR-10**    For CIFAR-10, we trained a model in pixel space without any VAE latent encoders. The model was trained without class conditioning, based on the UNet+(Song et al., 2021) backbone. We trained the model for 950K steps with a global batch size of 256. Non-leaky data augmentation (Karras et al., 2022) is also applied, excluding vertical flipping and rotation. Further details of the experimental settings can be found in Table 6.

Table 6: Experimental Settings

| Dataset | ImageNet $256 \times 256$ | | | | CIFAR-10 |
|---|---|---|---|---|---|
| Preprocessor | SD-VAE (Rombach et al., 2022) | | | | Identity |
| Input size | $32 \times 32 \times 4$ | | | | $32 \times 32 \times 3$ |
| Backbone | DiT-B/4 | DiT-B/2 | DiT-L/2 | DiT-XL/2 | UNet+ (Song & Ermon, 2019b) |
| Params (M) | 131 | 131 | 459 | 676 | 56 |
| Depth | 12 | 12 | 24 | 28 | - |
| Hidden dim | 768 | 768 | 1024 | 1152 | - |
| Heads | 12 | 12 | 16 | 16 | - |
| Patch size | $4 \times 4$ | $2\times2$ | $2\times2$ | $2\times2$ | $1\times1$ |
| Dropout | 0.0 | | | | 0.2 |
| Self-distillation | $\mathcal{L}_{\text{iSD-U}}$ | | | | $\mathcal{L}_{\text{iSD}}$ |
| Joint training | Enabled | | | | Enabled |
| JVP | Approximation: Eq. 17 | | | | `torch.func.jvp` |
| $\epsilon$ | 0.005 | | | | - |
| $p$ | 1.0 | | | | 0.75 |
| $\eta$ | 0.01 | | | | 0.01 |
| Pre-CFG $\omega$ | 1.5 | | | | - |
| Training steps | 800K | | | | 950K |
| Batch size | 256 | | | | 256 |
| Label dropout | 0.1 | | | | - |
| Optimizer | AdamW (Loshchilov & Hutter, 2019) | | | | AdamW |
| Learning rate | 1e-4 | | | | 1e-3 |
| LR Scheduler | Constant | | | | Linear Warmup |
| $\beta_1$ | 0.9 | | | | 0.9 |
| $\beta_2$ | 0.999 | | | | 0.999 |
| Weight decay | 0 | | | | 0 |
| EMA decay | 0.99995 | | | | 0.99995 |

## C.3 ADDITIONAL QUANTITATIVE RESULTS

Table 7: **Quantitative results across design choices**. The numeric entries in the header denote Post-CFG scales.

| Loss | Interp. | JVP | Steps | FID↓ | 1.5 | 3.0 | 7.0 | 10.0 |
|---|---|---|---|---|---|---|---|---|
| $\mathcal{L}_{\text{CT}}$ | Linear | Exact | 2 | 76.39 | 50.45 | 39.33 | 68.82 | 85.87 |
| | | | 4 | 73.57 | 47.93 | 25.80 | 38.19 | 55.88 |
| $\mathcal{L}_{\text{CT}}$ | Trig | Exact | 2 | 103.35 | 72.42 | 39.52 | 41.27 | 59.16 |
| | | | 4 | 83.51 | 52.83 | 21.33 | 17.48 | 26.43 |
| $\mathcal{L}_{\text{iSD}}$ | Linear | Exact | 2 | 118.17 | 98.77 | 77.75 | 89.02 | 100.61 |
| | | | 4 | 120.68 | 101.23 | 73.45 | 64.55 | 71.30 |
| $\mathcal{L}_{\text{iSD}}$ | Trig | Exact | 2 | 115.93 | 86.46 | 51.85 | 51.34 | 66.79 |
| | | | 4 | 100.35 | 71.40 | 35.41 | 26.54 | 34.58 |
| $\mathcal{L}_{\text{CT}}$ | Linear | Approx | 2 | 65.98 | 40.97 | 33.33 | 67.99 | 85.69 |
| | | | 4 | 62.70 | 37.44 | 19.25 | 36.41 | 58.07 |
| $\mathcal{L}_{\text{iSD}}$ | Linear | Approx | 2 | 112.39 | 90.26 | 69.81 | 87.50 | 101.42 |
| | | | 4 | 113.42 | 91.58 | 62.71 | 58.54 | 68.42 |
| $\mathcal{L}_{\text{iSD-U}}$ | Linear | Approx | 2 | 75.57 | 55.58 | 53.88 | 86.95 | 99.97 |
| | | | 4 | 77.55 | 57.44 | 43.86 | 57.20 | 71.43 |
| $\mathcal{L}_{\text{iSD-U}}$ | Trig | Approx | 2 | 66.63 | 41.84 | 27.99 | 47.04 | 66.25 |
| | | | 4 | 60.76 | 36.01 | 19.40 | 25.67 | 38.70 |

Table 8: **Quantitative results over training steps** (2-NFE). The numeric entries in the header denote training steps.

| Loss | Interp. | JVP | Arch. | Pre-CFG $\omega$ | 10K | 100K | 200K | 300K | 400K |
|---|---|---|---|---|---|---|---|---|---|
| $\mathcal{L}_{\text{CT}}$ | Linear | Exact | DiT-B/4 | - | 329.02 | 103.73 | 88.08 | 79.44 | 76.39 |
| $\mathcal{L}_{\text{CT}}$ | Trig | Exact | DiT-B/4 | - | 433.04 | 135.31 | 118.56 | 109.97 | 103.35 |
| $\mathcal{L}_{\text{iSD}}$ | Linear | Exact | DiT-B/4 | - | 392.22 | 156.94 | 136.04 | 127.76 | 118.17 |
| $\mathcal{L}_{\text{iSD}}$ | Trig | Exact | DiT-B/4 | - | 381.09 | 151.88 | 132.77 | 124.58 | 115.93 |
| $\mathcal{L}_{\text{CT}}$ | Linear | Approx | DiT-B/4 | - | 413.13 | 97.59 | 75.87 | 69.51 | 65.98 |
| $\mathcal{L}_{\text{iSD}}$ | Linear | Approx | DiT-B/4 | - | 425.42 | 151.88 | 129.41 | 119.42 | 112.39 |
| $\mathcal{L}_{\text{iSD-U}}$ | Linear | Approx | DiT-B/4 | 1.5 | 425.40 | 124.84 | 92.80 | 83.74 | 75.57 |
| $\mathcal{L}_{\text{iSD-U}}$ | Trig | Approx | DiT-B/4 | 1.5 | 414.77 | 116.95 | 87.89 | 75.59 | 66.63 |
| $\mathcal{L}_{\text{iSD-C}}$ | Linear | Approx | DiT-B/4 | 1.5 | 380.05 | 169.04 | 130.94 | 113.43 | 102.21 |
| $\mathcal{L}_{\text{iSD-C}}$ | Linear | Approx | DiT-B/4 | 3.0 | 379.94 | 202.94 | 122.01 | 101.64 | 91.38 |
| $\mathcal{L}_{\text{iSD-C}}$ | Trig | Approx | DiT-B/4 | 1.5 | 413.45 | 167.09 | 136.24 | 122.36 | 114.16 |
| $\mathcal{L}_{\text{iSD-C}}$ | Trig | Approx | DiT-B/4 | 3.0 | 412.07 | 150.26 | 107.73 | 90.49 | 82.59 |
| $\mathcal{L}_{\text{iSD-C}}$ | Trig | Approx | DiT-B/4 | 7.0 | 412.92 | 195.38 | 129.86 | 113.47 | 112.64 |
| $\mathcal{L}_{\text{iSD-U}}$ | Trig | Approx | DiT-B/2 | 1.5 | 384.36 | 103.10 | 69.50 | 57.36 | 50.58 |
| $\mathcal{L}_{\text{iSD-U}}$ | Trig | Approx | DiT-XL/2 | 1.5 | 410.63 | 89.34 | 57.63 | 44.54 | 38.50 |

# D QUALITATIVE RESULTS

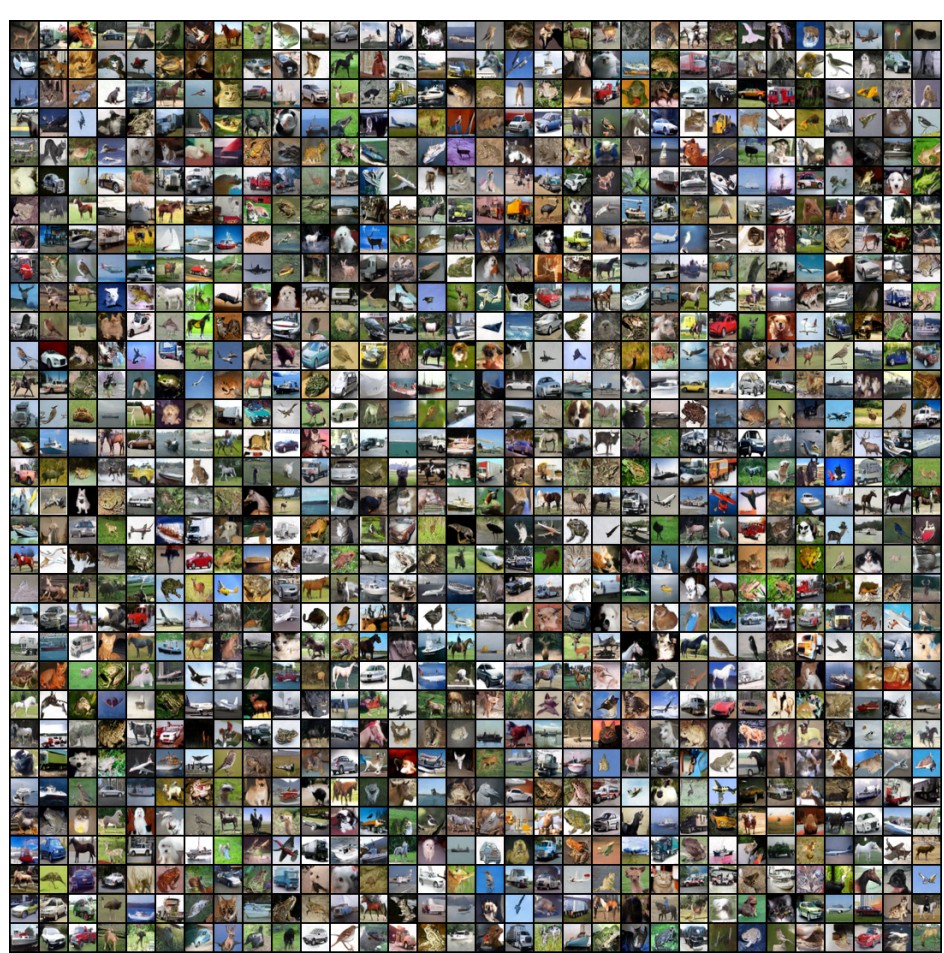

Figure 4: One-step samples from the vanilla iSD on CIFAR-10 (FID 3.64)

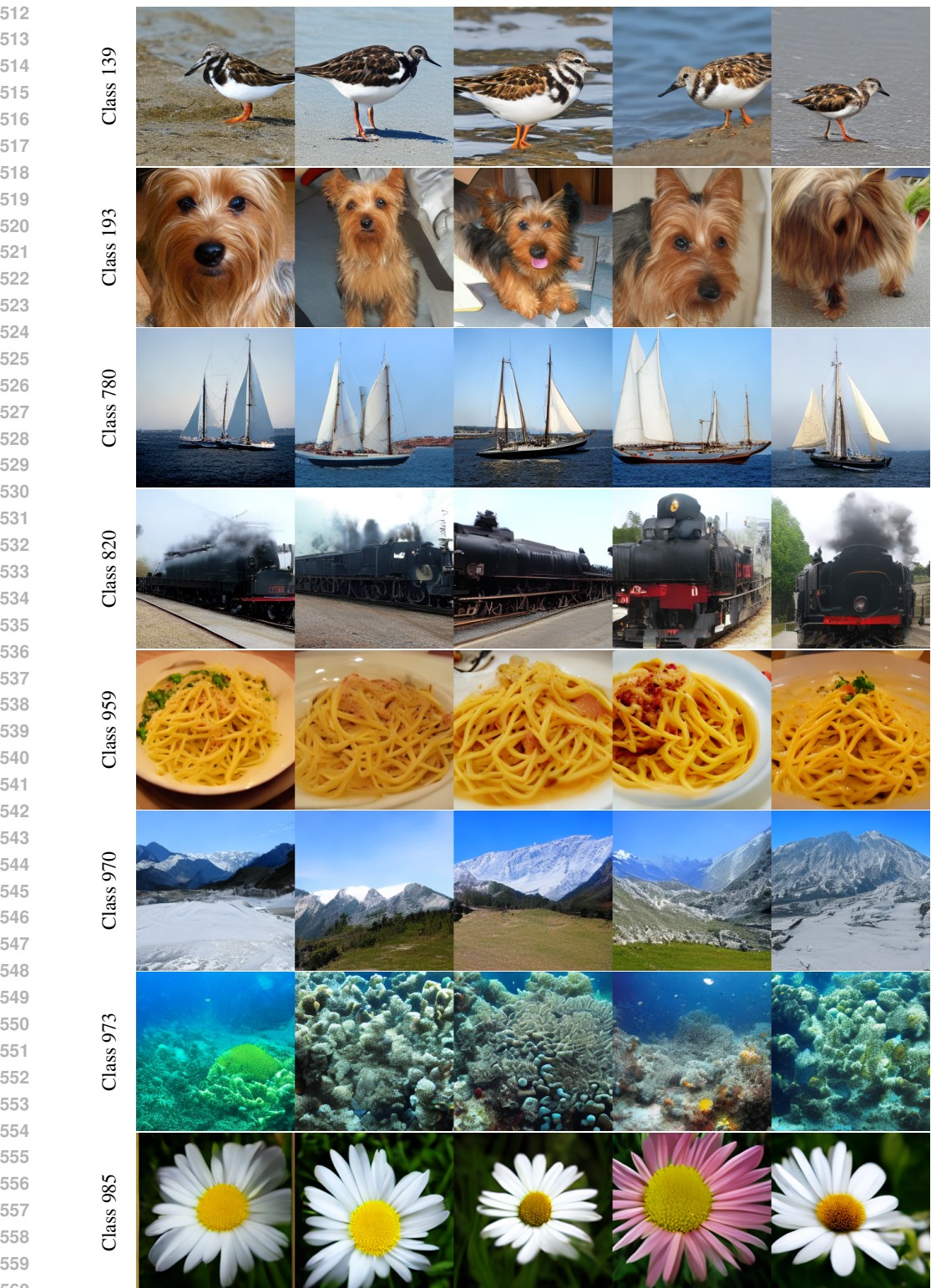

Figure 5: Class-level samples generated by iSD-U with four-step sampling on ImageNet $256 \times 256$

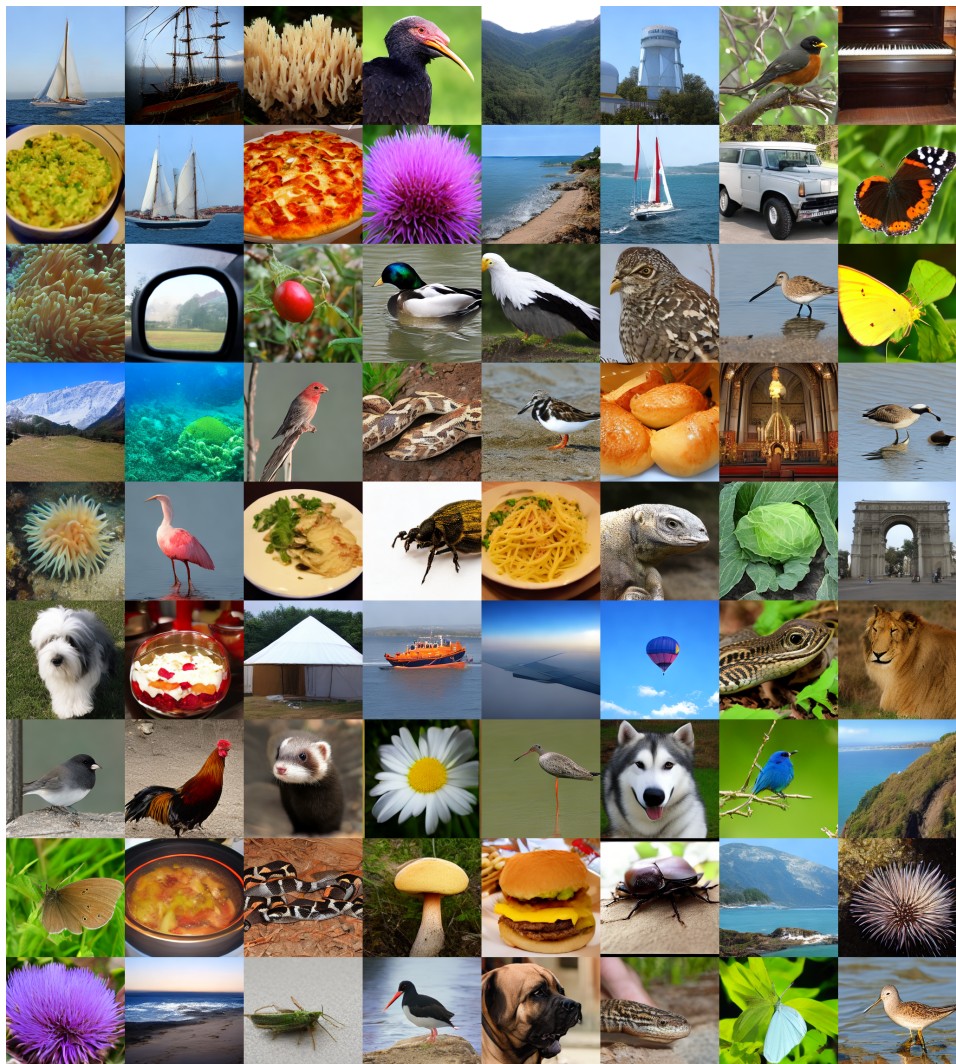

Figure 6: Four-step samples from the iSD-U on ImageNet $256 \times 256$

# E    USE OF LARGE LANGUAGE MODELS

OpenAI's ChatGPT was used to polish writing during preparation of this work. All text generated by the tool was reviewed and revised by the authors.

