# OpenReview forum: "Alleviating Suboptimality of Flow Maps with Improved Self-Distillation Guidance"
_ICLR.cc/2026/Conference — ICLR 2026 Conference Withdrawn Submission_

### Official Review · Reviewer_g7CR · 2025-10-27

**Soundness:** 2
**Presentation:** 2
**Contribution:** 2
**Rating:** 2
**Confidence:** 3

**Summary:**

The paper analyzes the weaknesses and instabilities of consistency and flow-map models, proposing an improved self-distillation objective to simplify the training procedure and improving stability. The method is validated on common image generation benchmarks.

**Strengths:**

The paper provides an analysis of why consistency training can be unstable, and proposes a grounded method to overcome such an instability.

**Weaknesses:**

I think the paper can benefit from a more clear writing. I found it hard to understand the contribution of the proposed improvements from the experimental section.
In particular, the actual contributions of the paper are reported in section 4.2. From my understanding, the main contribution is in the reformulation of the target for the consistency/flow-map loss, where the authors use the use the instantaneous velocity learned by the online model in the target, rather than the conditional velocity. I think an ablation on how this new target works compared to the one from eq.13 could help understanding the relevance of the contribution, keeping the remaining settings unchanged.

The Adaptive Weighting is not novel, and is the one used in MeanFlow and similar to pseudo-huber loss when p=0.5, so should not be mentioned as a contribution.

The use of the relaxed objective is not new and is already used in Inductive Moment Matching [1], MeanFLow, and other flow-map-based methods. The fact that using the Flow Matching Loss improves stability is also not new, and use in for example in FACM [2], even though the method is relatively recent and could be considered as concurrent work, but should definitely be added to the references.

The ablation part is unclear to me (see questions), and the experimental results on Imagenet are not competitive with state of the art.

The results on CIFAR-10 are actually relatively close to SOTA, which is generally < 3 FID. The improvement is substantial compared to [3], but a more systematic analysis using the same settings as in [3] and highlighting what contributes to the improvement could help appreciating the significance of the contribution.

## References
[1] Zhou, Linqi, Stefano Ermon, and Jiaming Song. "Inductive Moment Matching."

[2] Peng, Yansong, et al. "Flow-anchored consistency models."

[3] Boffi, Nicholas M., Michael S. Albergo, and Eric Vanden-Eijnden. "How to build a consistency model: Learning flow maps via self-distillation."

**Questions:**

- 1) In Table 3, the experiments start with a baseline. What is the baseline?
- 2) The second and third entry with CFM and CT relaxation are respectively the addition of L_CFM and the s-conditioning, correct?
- 3) What is the exactly the addition of self-distillation? Does it refer to Eq 14 or to the addition of CFG with weight w=1.5?
- 4) Why would simply switching from the linear interpolation to the TrigFlow kernel bring such a big boost in performance?
- 5) Are your CIFAR-10 result obtained with CFG? It's not entirely clear to me what the final configuration is.
- 6) Are the authors using a pretrained model or are the weights initialized at random?

---

### Official Review · Reviewer_5saY · 2025-10-31

**Soundness:** 3
**Presentation:** 3
**Contribution:** 2
**Rating:** 4
**Confidence:** 4

**Summary:**

This paper presents Improved Self-Distillation Guidance (iSD), a new framework for training Consistency Models (CMs) and Flow Matching (FM) models that aims to mitigate the suboptimality and instability issues inherent in existing methods. The authors first unify CMs and FMs under a generalized flow map formulation, identifying that conventional training objectives optimize conditional rather than marginal velocity fields—leading to unstable or irreproducible training. Building on this insight, they propose iSD, which relaxes the time constraint
$s=0 \rightarrow s < t$ and introduces self-distillation guided by marginal velocities. This modification improves convergence stability and reproducibility without requiring pretraining or heuristic regularization. Empirically, iSD achieves competitive or superior results on CIFAR-10 and ImageNet-256 with few-step sampling, outperforming prior self-distillation and shortcut-based methods in both FID and stability.

**Strengths:**

1. The paper provides a conceptually clear and theoretically grounded reformulation of consistency training through the generalized flow map framework, offering a rigorous explanation of why existing self-distillation methods underperform.
2. The iSD objective is a simple yet elegant modification that improves both the stability and reproducibility of CM training while reducing dependence on heuristic regularizers.
3. Empirical results are comprehensive and reproducible, showing substantial gains in both FID and standard deviation across runs—addressing a long-standing criticism of CM reproducibility.
4. The introduction of Pre-CFG guidance as an integrated training mechanism is a practically relevant addition that aligns well with modern diffusion model practices.

**Weaknesses:**

1. While the theoretical motivation is solid, the derivation and intuition behind marginal vs. conditional velocity mismatch could be explained more intuitively; the connection to variance reduction or stochastic gradient alignment is only briefly mentioned.
2. The computational trade-offs of iSD—especially the approximate JVP computation—are not quantified in FLOPs or runtime, leaving uncertainty about scalability to higher-resolution datasets.
3. Evaluation is primarily limited to ImageNet-256 and CIFAR-10; demonstrating results on higher-resolution tasks (e.g., text-to-image or unconditional generation) would strengthen the paper’s generality claims.
4. Although the method improves reproducibility, ablation studies on the impact of each design choice (e.g., time relaxation vs. marginal guidance) are limited.
5. The paper’s theoretical component stops short of a formal convergence guarantee; although the empirical stability results are convincing, a deeper analysis of training dynamics (e.g., curvature of loss landscape) would enhance credibility.

**Questions:**

1. Could the authors provide more intuition on how marginal velocity alignment alleviates suboptimality in practice? Is this equivalent to a bias correction in gradient estimation?
2. How sensitive is iSD to the choice of the relaxation schedule for $s<t$? Would adaptive or learned schedules further improve convergence?
3. What is the computational overhead of the approximate JVP calculation, and does it affect inference latency?

---

### Official Review · Reviewer_Momx · 2025-11-01

**Soundness:** 3
**Presentation:** 3
**Contribution:** 2
**Rating:** 4
**Confidence:** 3

**Summary:**

This paper proposes a generalized flow map framework that unifies recent consistency-based generative models and identifies two root causes of their suboptimality and instability: (i) lack of marginal velocity guidance and (ii) fixed time conditioning $(s = 0)$. To address these, the authors introduce improved Self-Distillation (iSD), which combines self-distilled marginal velocity guidance with relaxed time conditioning $(s < t)$, JVP approximation, and classifier-free guidance (CFG). The method trains from scratch without heuristics and achieves competitive 4-step FID of 11.06 on ImageNet 256×256 with high reproducibility. The work provides theoretical analysis of convergence gaps and empirical validation across design choices.

**Strengths:**

1. Generalized flow map (Eq. 6) cleanly subsumes prior methods (Tab. 1); Propositions 3.3–3.4 rigorously show suboptimality/instability of direct and consistency training.
2. The method builds on DiT and avoids distillation from external teachers. Comparisons to UCGM, MeanFlow, Shortcut Model, and iCT are fair and comprehensive.

**Weaknesses:**

1. All main results are on class-conditional ImageNet 256x256. There is no demonstration on unconditional generation, higher resolutions (512x512), or other modalities (e.g., text-to-image). This limits claims of generality.
2. While stable, iSD requires ~800K iterations and joint optimization of $L_{CFM} + L_{SD}$, which is slower than one-stage consistency training (e.g., UCGM). No wall-clock time or GPU-hour comparison is provided.
3. iSD-U (Pre-CFG) is empirically best but relaxes $s = t$, violating the marginal flow map convergence proof. The paper acknowledges this but does not quantify the practical impact of this compromise.

**Questions:**

Refer to the weaknesses.

---

### Note · Authors · 2025-11-12

**Comment:**

Thank you for your reviews. Based on the comments, we have decided to withdraw our submission. We truly appreciate the constructive suggestions, and we will incorporate them to improve the work for future publication.

**Withdrawal Confirmation:**

I have read and agree with the venue's withdrawal policy on behalf of myself and my co-authors.